# Stable Asynchrony: Variance-Controlled Off-Policy RL for LLMs

**Luke J. Huang** [1]  **Zhuoyang Zhang** [1]  **Qinghao Hu** [1]  **Shang Yang** [1]  **Song Han** [1,2]

## Abstract

Asynchronous reinforcement learning has become increasingly central to scaling LLM post-training, delivering major throughput gains by decoupling rollout generation from policy updates. However, widely used policy-gradient objectives such as REINFORCE and GRPO suffer under high asynchrony: stale rollouts produce heavy-tailed importance weights, so a small number of trajectories dominate updates and the policy-gradient estimator becomes markedly *higher variance*. Through systematic analysis on math, reasoning, and tool-use benchmarks, we establish that this increasing variance is reliably predicted by collapsing effective sample size (ESS), which prior stabilization methods largely fail to address. Motivated by this diagnosis, we introduce **V**ariance **C**ontrolled **P**olicy **O**ptimization (**VCPO**), a method that (i) dynamically scales the learning rate with ESS to dampen unreliable updates and (ii) applies a closed-form minimum-variance baseline for off-policy settings, without a critic model and adding minimal overhead. Empirically, across math and general reasoning benchmarks, this enables robustly stable asynchronous training compared to previous stabilization and algorithmic methods, even in highly off-policy regimes (128 steps off-policy). In a long-horizon, tool-use task, VCPO matches synchronous performance while delivering a 2.5× speedup in training time. Code is available at: https://github.com/mit-han-lab/vcpo

## 1. Introduction

Policy-gradient (PG) methods directly optimize a parameterized policy from sampled trajectories and have become the dominant approach for improving reasoning in large language models. Common objectives include Proximal Policy Optimization (PPO) (Schulman et al., 2017) as well as REINFORCE (Williams, 1992) and its recent variants such as GRPO (Shao et al., 2024).

These methods are typically implemented in an on-policy or near on-policy regime, where data generation and learning must occur sequentially after each other. This coupling becomes a major systems bottleneck for long-horizon tasks: the distribution of rollout lengths exhibit a long tail (Hu et al., 2025), leading to pipeline bubbles and poor GPU utilization in synchronous RL training.

To improve end-to-end training speed, asynchronous RL fully decouples rollout generation from learning via asynchronous, pipelined training. Prior works such as asynchronous RLHF (Noukhovitch et al., 2024) AReaL (Fu et al., 2025), LlamaRL (Wu et al., 2025), and PipelineRL (Piché et al., 2025) have demonstrated up to 3× end-to-end speedups. However, this efficiency comes at the cost: training becomes strongly off-policy as the sampler generates trajectories with a lagging (stale) behavior policy. While importance sampling offers theoretically unbiased correction estimator, it fails to prevent learning degradation or even collapse in practice. Previous works have largely addressed this by suppressing outlier importance sampling ratios via masking, clipping, or other whitening and reshaping techniques, but in practice we find that these methods are insufficient for highly asynchronous training.

In contrast, we identify a deeper optimization failure mode: asynchronous RL training does not *explicitly* account for the variance of the policy-gradient estimator itself. To our knowledge, we are first to establish that training instability is reliably predicted by a collapsing effective sample size (ESS) (Kish, 1965) of the importance-weighted gradient estimate, validating across math, general reasoning, and tool-use benchmarks.

Motivated by this insight, we propose **V**ariance **C**ontrolled **P**olicy **O**ptimization (VCPO): a method with explicit variance-targeted controls that stabilize policy-gradient objectives for asynchronous RL training. VCPO consists of two complementary components: (i) an *effective sample size guided step scaling* that down-weights unreliable updates and (ii) a *closed-form minimum-variance off-policy reward baseline* based on per-trajectory gradient norms, avoiding

---

[1]MIT [2]NVIDIA. Correspondence to: Luke Huang <lukh23@mit.edu>, Song Han <songhan@mit.edu>.

*Proceedings of the 43rd International Conference on Machine Learning*, Seoul, South Korea. PMLR 306, 2026. Copyright 2026 by the author(s).

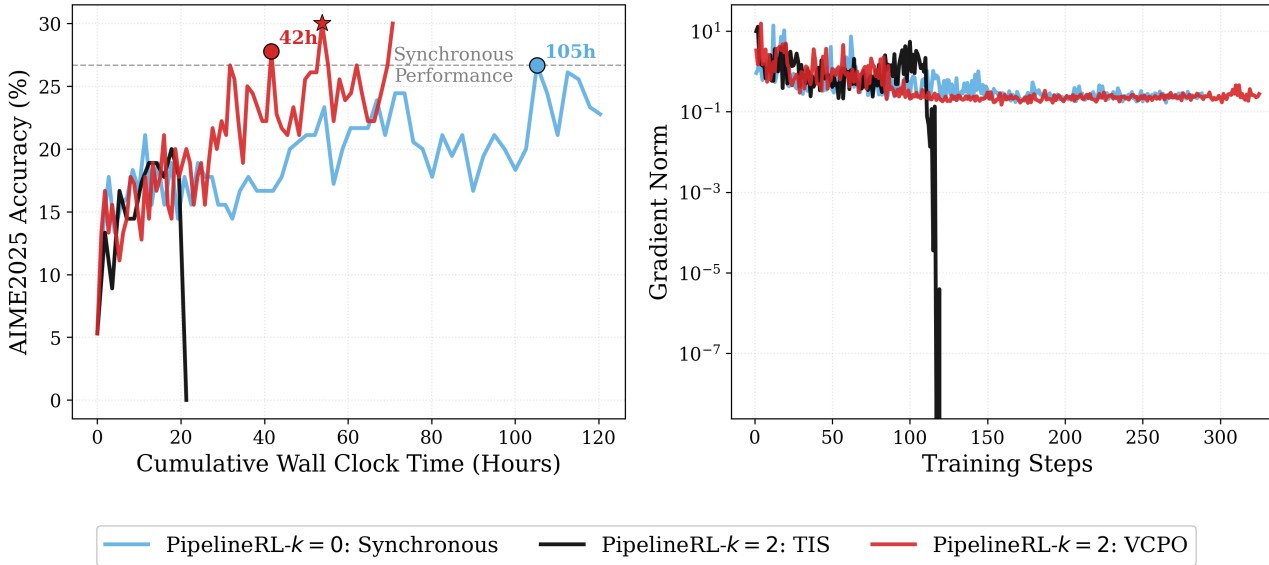

*Figure 1.* Long-context, tool-integrated multi-turn RL comparing synchronous training ($k$=0) to a two-step policy lag ($k$=2) with VCPO versus sequence-level truncated importance sampling (TIS). Left: AIME-2025 validation accuracy vs. cumulative wall-clock time. VCPO matches the best synchronous accuracy 2.5× faster ($\approx$42h vs. $\approx$105h) and continues to improve thereafter. Right: Gradient norm vs. training steps. The TIS run shows an instability characterized by a brief gradient-norm spike followed by rapid collapse.

training an auxiliary critic or value model. To make (ii) practical at scale, we present the first scalable implementation for *exact* per-example gradient-norm computation with negligible runtime overhead, and leverage these statistics to build a low-overhead, gradient-aware baseline for variance reduction.

Across highly asynchronous training regimes and across mathematical reasoning, general reasoning, and tool-use tasks, VCPO remains robust and effective from 1.5B to 7B models, outperforming a broad suite of baselines spanning masking/clipping stabilizers and algorithmic variants up to 128× steps of off-policyness. When applied to long-context multi-turn training, VCPO reduces end-to-end training time by 2.5× while matching synchronous performance.

**Conflict of Interest Disclosure.** The authors declare no conflicts of interest.

## 2. Background

### 2.1. Reinforcement Learning for LLMs

We consider finetuning a pretrained language model policy $\pi_\theta$ to maximize a scalar reward $R(x, y)$ that scores a completion $y = (y_1, \ldots, y_T)$ for a prompt $x$ (e.g., from a reward model or task-specific evaluator). Under an autoregressive policy, the likelihood of generating $y$ is

$$\pi_\theta(y \mid x) = \prod_{t=1}^{T} \pi_\theta(y_t \mid x, y_{<t}).$$

Let $\mathcal{D}$ denote a dataset (distribution) of prompts. RL optimizes the expected reward

$$J(\theta) = \mathbb{E}_{x \sim \mathcal{D}, \, y \sim \pi_\theta(\cdot \mid x)}[R(x, y)].$$

**On-Policy Gradient Estimator**. In On-Policy RL, we estimate $\nabla_\theta J(\theta)$ with sequences $y$ sampled from the policy $\pi_\theta$ itself. This leads to the REINFORCE gradient ([Williams, 1992](#))

$$\nabla_\theta J(\theta) = \mathbb{E}_{x \sim \mathcal{D}, \, y \sim \pi_\theta}[R(x, y) \, \nabla_\theta \log \pi_\theta(y \mid x)]$$

$$\nabla_\theta \log \pi_\theta(y \mid x) = \sum_{t=1}^{T} \nabla_\theta \log \pi_\theta(y_t \mid x, y_{<t})$$

which motivates a surrogate loss of

$$\mathcal{L}_{\text{on-policy}}(\theta) = -\mathbb{E}_{x \sim \mathcal{D}, \, y \sim \pi_\theta(\cdot \mid x)}[R(x, y) \, \log \pi_\theta(y \mid x)] \tag{1}$$

A standard variance reduction technique is to subtract a baseline that may depend on the state $s_t = (x, y_{<t})$, but not on the current action/token $y_t$:

$$\nabla_\theta J(\theta) = \mathbb{E}_{x, \, y \sim \pi_\theta}\left[\sum_{t=1}^{T} A_t \, \nabla_\theta \log \pi_\theta(y_t \mid x, y_{<t})\right],$$

where an advantage-like term can be written as

$$A_t = R(s_t, y_t) - b(s_t), \qquad b(s_t) = b(x, y_{<t}).$$

In actor–critic methods such as PPO, the baseline is a learned value function $b(s_t) = V_\phi(s_t)$ and advantages $A_t$

are computed via GAE. Critic-free methods such as GRPO instead use structured baselines (e.g., per-prompt group normalization), avoiding the overhead of training and serving a value model. In this paper, we focus on critic-free, REINFORCE-style updates, which are widely used for LLM reasoning post-training due to their lower resource requirements.

**Off-Policy Gradient Estimator**. In practical LLM RL systems, rollout generation and training are inevitably mismatched, causing training to be off-policy. This mismatch can arise even without explicit asynchronous training due to numerical issues (Qi et al., 2025; Xi et al., 2026), kernel and hardware issues (Liu et al., 2025a), or unavoidable differences in inference and training implementations (He, 2025). We therefore distinguish the *sampler* policy $\mu$ used to generate trajectories from the current *learner* policy $\pi_\theta$ being optimized.

Importance Sampling (IS) corrects this mismatch. For a prompt $x$ and completion $y$ consisting of tokens $y_1, \cdots, y_T$, define the (sequence-level) IS ratio

$$w(x, y) \triangleq \frac{\pi_\theta(y \mid x)}{\mu(y \mid x)} = \prod_{t=1}^{T} \frac{\pi_\theta(y_t \mid y_1^{t-1}x)}{\mu(y_t \mid y_1^{t-1}x)} \quad (2)$$

With importance sampling, Eq (1) can be expressed as an expectation relative to the *sampler policy*, which leads to the unbiased off-policy surrogate loss

$$\mathcal{L}_{\text{off-policy}}(\theta) = -\mathbb{E}_{x \sim \mathcal{D}, \, y \sim \mu(\cdot \mid x)} \big[ w(x, y) \, A(x, y) \\ \log \pi_\theta(y \mid x) \big]. \quad (3)$$

where $A(x, y)$ denotes an advantage-like signal (e.g., $R - b$ or token-level advantages).

## 2.2. Why Effective Sample Size?

In practice, the core challenge of importance sampling is not bias but *finite-sample variance*. Known as the *Curse of the Horizon* (Liu et al., 2018; 2020), the product structure in (2) makes $w(x, y)$ highly sensitive to small per-token probability shifts, so the resulting weights can become heavy-tailed and a few samples may dominate each update

To quantify this weight degeneracy, given per-sample gradient contributions $g_i$ and importance weights $w_i$, define standardized weights $\tilde{w}_i = w_i / \sum_{j=1}^{B} w_j$ and the corresponding weighted estimator $\hat{g} = \sum_{i=1}^{B} \tilde{w}_i g_i$. A common diagnostic is the **effective sample size** (ESS) (Kong, 1992; Kong et al., 1994):

$$\text{ESS} \triangleq \frac{\left( \sum_{i=1}^{B} w_i \right)^2}{\sum_{i=1}^{B} w_i^2} = \frac{1}{\sum_{i=1}^{B} \tilde{w}_i^2} \in [1, B].$$

ESS measures how many samples effectively contribute to the weighted estimate: if weights are nearly uniform then

ESS $\approx B$, while if a few weights dominate then ESS $\ll B$. Moreover, under mild conditions (e.g., weak correlation between $g_i$ and $w_i$), we have

$$\text{Var}(\hat{g}) \approx \left( \sum_{i=1}^{B} \tilde{w}_i^2 \right) \text{Var}(g) = \frac{1}{\text{ESS}} \text{Var}(g),$$

Intuitively, this means the variance of off-policy gradient estimates only matches the on-policy averaging rate with $B$ replaced by ESS. Thus, when ESS collapses, gradient estimates become substantially higher variance, leading to high step-to-step volatility and KL. For this reason, practical RL systems and algorithms often apply clipping (Schulman et al., 2017), truncation (Espeholt et al., 2018), or masking which limit the influence of extreme weights, at the cost of introducing bias relative to the exact IS correction.

## 2.3. Prior RL stabilization methods

While prior methods have been proposed to stabilize LLM RL in on-policy or near-on-policy settings, we discuss these works as they closely related to asynchronous training's highly off-policy issues. These approaches broadly fall into three categories: (1) masking/clipping to suppress outlier updates, (2) algorithmic changes to the estimator/objective, and (3) system-side changes that reduce sampler–learner mismatch.

**Masking / clipping mechanisms.** Similar to truncation and clipping mechanisms proposed by previous works (Espeholt et al., 2018; Schulman et al., 2017), recent work has propsed masking or truncating samples with outlier importance-weight ratios to prevent them from dominating updates.

Token or sequence-level TRUNCATED IMPORTANCE SAMPLING (TIS) constrains the IS ratio maximum. TIS has been used in large-scale asynchronous systems such as LLAMARL (Wu et al., 2025) and AREAL (Fu et al., 2025) as well as in near on-policy RL to correct learner–sampler mismatch in modern LLM RL training stacks (Yao et al., 2025a;b)

Closely related is token or sequence-level MASKED IMPORTANCE SAMPLING (MIS), which discard away updates with extreme IS ratios. ICEPOP proposed token-level MIS to improve stability in MoE RL (Zhao et al., 2025) while other works motivate sequence-level MIS due training–inference mismatch analyses (Liu et al., 2025a). Notably, DEEPSEEK-V3.2 uses masking sequences above a geometric-mean threshold and with negative advantages, illustrating a practical masking rule used at scale (DeepSeek-AI et al., 2025).

Finally, M2PO iteratively drops token-level losses until a second-moment proxy of importance weights falls below a threshold (Zheng et al., 2025b;c).

**Algorithmic changes.** GSPO improves training stability in MoE RL, defining importance ratios as the geometric mean of the token-level IS ratios over the sequence and performs sequence-level clipping/optimization (Zheng et al., 2025a).

Minimum-variance baselines have also been studied as an algorithmic way to reduce policy-gradient variance. Zhao et al. (Zhao et al., 2013) derive a closed-form optimal baseline for importance-weighted policy-gradient estimators. In contemporary LLM-RL, OPTIMAL TOKEN BASELINE (OTB) derives a token-level optimal baseline for long-horizon LLM-RL and an off-policy/TIS extension, and proposes a logit-gradient proxy for score-function norms to avoid extra backward passes (Li et al., 2026). However, prior work did not provide a way to compute the exact gradient-norm terms required by the minimum-variance baseline cheaply in large-scale LLM RL, which has been a significant barrier to using exact minimum-variance baselines in this setting.

**System-side changes.** A complementary direction reduces mismatch via numerical and systems alignment. Switching from BF16 to FP16 can reduce train–inference numerical inconsistencies and improve stability (Qi et al., 2025), while Rollout Routing Replay (R3) stabilizes MoE RL by replaying inference-time expert routing during training (Ma et al., 2025). These mitigations are orthogonal to variance-aware optimization and can be combined with our approach.

### 2.4. Asynchronous RL

In this paper, we adopt PIPELINERL-$k$ (Piché et al., 2025) for our asynchronous training setup. At learner update $t$, the trainer optimizes the current policy $\pi_{\theta_t}$ using trajectories sampled by a (possibly stale) behavior policy $\mu = \pi_{\theta_{t'}}$, where $k$ bounds the *policy lag* $t - t'$ (with $k = 0$ recovering the synchronous/on-policy regime). This induces off-policy updates whose severity increases with $k$. We also follow PipelineRL's *in-flight* weight-update scheme, where sampling continues while weights are being updated. As a result, a single trajectory may span multiple policy versions.

Throughout this paper, we use *sequence-level* importance sampling to correct for this lag, which is the theoretically grounded importance sampling for sequence-level policy gradients, and optimize the importance-weighted surrogate objective in (3).

## 3. VCPO: Robust, Scalable Asynchronous RL

We develop VCPO, a variance-controlled optimization framework for asynchronous RL. We first diagnose collapse through effective sample size (ESS), showing that dominated, high-variance updates precede KL and reward collapse. We then derive a ESS-aware learning rate scaling rule and the variance-minimizing off-policy baseline (OPOB) for importance-weighted policy gradients. Finally,

we give a single-backward implementation and combine these components into VCPO.

### 3.1. Why does Asynchronous RL collapse?

To understand training collapse under asynchrony, we run controlled math-RL diagnostics across two model families: Qwen2.5-7B Base trained on MATH (Hendrycks et al., 2021), a dataset of competition-style mathematics problems scored by exact-match final-answer correctness, and Llama3.1-8B Instruct trained on ORZ Math. Both runs are evaluated on 500 held-out MATH problems (MATH-500), with the Llama training setup and hyperparameters detailed in Appendix F.3. We log three signals over training: (i) the KL divergence between the rollout policy and the current learner policy (policy lag), (ii) reward / validation accuracy, and (iii) the *ESS ratio*, i.e., the effective sample size computed from sequence-level importance weights and normalized by the minibatch size.

**Baseline choice for diagnostics.** To isolate the effect of policy lag, we adopt a strong masking/clipping baseline for these didactic collapse experiments. Based on the masking and clipping sweeps in Appendix E.2, we find that *sequence-level* truncated importance sampling (TIS) with a high threshold ($c = 8.0$) delays collapse the longest among masking/clipping methods. We therefore use this setting as our default off-policy correction when diagnosing collapse.

**Optimizer and KL regularization.** For all experiments we use AdamW with weight decay (see Appendix F.1 for details). Following prior asynchronous (Fu et al., 2025; Wu et al., 2025) and synchronous (Yu et al., 2025; Shah et al., 2026) RL works, we do not incorporate explicit KL regularization loss as it constrains learning ability (corroborated by our own experiments in Appendix E.3).

**Training Collapse.** Figure 2 shows representative collapses under high staleness across two model families. As the ESS ratio collapses, updates become dominated by a few trajectories, leading to a KL explosion and an abrupt drop in training reward and validation accuracy. This supports our hypothesis that *high-variance, dominated updates* drives training collapse. For further evidence, see Appendix A for examples on math and reasoning tasks.

### 3.2. Effective Sample Size Guided Step-Size Scaling for Stable Updates

If asynchronous training fails due a subset of unreliable updates, a natural idea is to reduce the impact of those updates so that optimization remains robust under policy lag. A useful analogy comes from large-batch optimization: as batch size increases, gradient noise typically decreases (McCandlish et al., 2018), allowing larger stable learning rates. While SGD often follows the *linear scaling*

## Asynchronous RL Collapse on Math-500

*Figure 2.* **Sequence-Level TIS Collapse.** Qwen2.5-7B Base with 10-step off-policy (PIPELINERL-10) and Llama3.1-8B Instruct with 12-step off-policy (PIPELINERL-12) trained on math-reasoning data. In both runs, the ESS ratio first degrades and then collapses, leading to a spike in rollout–policy KL divergence and a sharp drop in both training reward and MATH-500 validation accuracy. See Appendix F.3 for hyperparameters and training details.

*rule* ($\eta \propto B$) (Goyal et al., 2017), a widely used heuristic for adaptive optimizers such as Adam/AdamW is the *square-root scaling law*: increasing minibatch size by a factor $k$ permits scaling the learning rate by $\sqrt{k}$ (equivalently, $\eta \propto \sqrt{B}$) (Krizhevsky, 2014; Hoffer et al., 2017; You et al., 2020).

In asynchronous off-policy RL, however, the *nominal* batch size $B$ can be misleading. As discussed in Section 2.2, importance weights may be heavy-tailed, so a batch of size $B$ can behave like a much smaller batch with only ESS effectively independent samples. In this regime, ESS is the relevant notion of effective batch size, and step-size selection should depend on ESS rather than on $B$.

Thus, motivated by the square-root scaling law $\eta \propto \sqrt{B}$, we propose learning-rate scaling via an effective batch-size rule

$$\eta_{\text{eff}} \propto \sqrt{\rho_{\text{ess}}}, \qquad \rho_{\text{ess}} \triangleq \frac{\text{ESS}}{B} \qquad (4)$$

where we call $\rho_{\text{ess}}$ the **effective sample size ratio**.

Because even on-policy/synchronous training can have $\rho_{\text{ess}}^{\text{on}} < 1$ in practice (e.g., due to inevitable training-sampling differences and finite-sample effects), we introduce an *empirical on-policy reference* $\rho_{\text{ess}}^{\text{on}} \triangleq \mathbb{E}[\rho_{\text{ess}} \mid \text{on-policy}]$ which is a **constant** estimated from 1 step of a on-policy run (or a running average of a few-steps)[1]. We then rescale the step

size by the relative reliability of the batch:

$$\eta_{\text{eff}} = \eta \cdot \sqrt{\frac{\rho_{\text{ess}}^{\text{off}}}{\rho_{\text{ess}}^{\text{on}}}} \qquad (5)$$

This scaling preserves the on-policy step scale when $\rho_{\text{ess}} \approx \rho_{\text{on}}$, and automatically shrinks updates as $\rho_{\text{ess}}$ collapses.

Thus when training is nearly on-policy and ESS $\approx B$ and $\eta_{\text{eff}} \approx \eta$; when a few samples dominate, ESS $\ll B$ and $\eta_{\text{eff}}$ shrinks like $\sqrt{\text{ESS}/B}$, damping the unstable, high-variance updates that coincide with KL/gradient spikes.

### 3.3. Minimum Variance Baselines for Off-Policy RL

Subtracting a baseline from the reward (or advantage) leaves the expected policy gradient unchanged but can substantially reduce variance. This control-variate view dates to reinforcement-comparison methods (Dayan, 1991) and was formalized for policy gradients (Greensmith et al., 2004). As discussed in Section 2.3, Zhao et al. (Zhao et al., 2013) previously derived the same importance-weighted optimal-baseline form, and OTB concurrently derived a related token-level LLM-RL baseline and off-policy/TIS extension (Li et al., 2026). In our work, we show an exact method of calculating the trajectory-level baseline with minimal overhead, allowing it to be efficiently integrated into asynchronous LLM RL and combined with our ESS-guided on-policy step.

---

[1] Empirically, $\rho_{\text{on}}$ is typically well-approximated by the first

step-size scaling.

**Setup.** Consider the off-policy gradient estimator with a scalar baseline $b$:

$$\hat{G}(b) = \frac{1}{B} \sum_{i=1}^{B} w_i (R_i - b) \nabla_\theta \log \pi_\theta(\tau_i) \quad (6)$$

where each sample $i$ contribute a score-gradient vector $g_i = \nabla_\theta \log \pi_\theta(\tau_i)$, a scalar reward/advantage $R_i$, and an importance ratio $w_i$.

**Optimal Off-policy Baseline.** As we derive in Appendix B, minimizing $\mathrm{Var}(\hat{G}(b))$ over scalar $b$ yields a closed-form solution for the off-policy optimal baseline (OPOB)

$$b^\star_{\mathrm{OPOB}} = \frac{\sum_{i=1}^{B} w_i^2 \|g_i\|^2 R_i}{\sum_{i=1}^{B} w_i^2 \|g_i\|^2}. \quad (7)$$

Compared to common group baselines (e.g., $b = \frac{1}{B} \sum_i R_i$), Eq. (7) shows that variance-optimal baselining in the off-policy regime depends on *both* (i) the importance weights $w_i$ and (ii) the gradient magnitudes $\|g_i\|^2$. Intuitively, samples that are both highly upweighted off-policy and induce large parameter changes dominate update variance and should therefore dominate the baseline.

**Connections to prior baselines.** Equation (7) is the importance-weighted optimal-baseline form derived by Zhao et al. (Zhao et al., 2013). When training is on-policy ($w_i = 1$), it reduces to the classic gradient-norm-weighted optimal baseline (Greensmith et al., 2004; Weaver & Tao, 2013). If, additionally, $\|g_i\|$ is approximately constant within a group, then $b^\star$ reduces to the group-mean reward baseline commonly used in practice. OTB concurrently derives a related token-level LLM-RL baseline and an off-policy/TIS extension (Li et al., 2026); our implementation differs by computing exact per-trajectory gradient norms for the full model in an asynchronous training system.

### 3.4. Efficient Baseline-Aware Gradient Computation

A naive gradient-aware baseline requires two backward passes: one to compute per-sample gradient norms and a second to apply the resulting baseline, roughly doubling backward compute time at scale. We instead implement it with a single backward pass, adding minimal wall-clock and memory overhead.

**Key observation: linearity of the baseline term.** Note that expanding the equation in (6) yields

$$\hat{G}(b) = \frac{1}{B} \underbrace{\sum_{i=1}^{B} w_i R_i g_i}_{\text{reward-weighted term}} - b \cdot \frac{1}{B} \underbrace{\sum_{i=1}^{B} w_i g_i}_{\text{score term}}$$
$$= \frac{1}{B} (G_R - b \, G_S). \quad (8)$$

---

**Algorithm 1** Single-backward accumulation with Off-Policy Optimal Baseline baseline

---

1: **Input:** Minibatch $\{(x_i, \tau_i, R_i, w_i)\}_{i=1}^{B}$, learner policy $\pi_\theta$
2: **Output:** Baseline-aware gradient estimate $\hat{G}(b^\star_{\mathrm{OPOB}})$ and scalar baseline $b^\star_{\mathrm{OPOB}}$
3: Initialize gradient buffers $G_R \leftarrow 0, G_S \leftarrow 0$
4: Initialize baseline numerators $N \leftarrow 0, D \leftarrow 0$
5: **for** $i = 1$ to $B$ **do**
6: $\quad \ell_i \leftarrow \log \pi_\theta(\tau_i \mid x_i)$
7: $\quad g_i \leftarrow \nabla_\theta \ell_i$ *(backprop)*
8: $\quad s_i \leftarrow \|g_i\|_2^2$
9: $\quad G_R \leftarrow G_R + (w_i R_i) g_i$
10: $\quad G_S \leftarrow G_S + w_i g_i$
11: $\quad N \leftarrow N + w_i^2 s_i R_i$
12: $\quad D \leftarrow D + w_i^2 s_i$
13: **end for**
14: $b^\star_{\mathrm{OPOB}} \leftarrow \mathrm{stopgrad}\left(\frac{N}{D+\varepsilon}\right)$
15: $\hat{G}(b^\star_{\mathrm{OPOB}}) \leftarrow \frac{1}{B} (G_R - b^\star_{\mathrm{OPOB}} G_S)$
16: **return** $\hat{G}(b^\star_{\mathrm{OPOB}}), b^\star_{\mathrm{OPOB}}$

---

where $G_R \triangleq \sum_{i=1}^{B} w_i R_i g_i$ and $G_S \triangleq \sum_{i=1}^{B} w_i g_i$. Thus, for any chosen $b$, the final gradient can be formed by combining two aggregated quantities $G_R$ and $G_S$.

**Score-gradient reweighting into two buffers.** We take advantage of this linearity by computing per-sample score gradients $g_i$ once and accumulate them into two gradient buffers: (i) $G_R$ a buffer that accumulates the reward/advantage-weighted score gradients $R_i w_i g_i$, and (ii) $G_S$ a score buffer that accumulates score gradients $w_i g_i$. Afterwards, we form the final update by a simple linear combination of these buffers following Eq. 8. This avoids any additional backward pass to incorporate the baseline term.

**Implementation.** We implement Algorithm 1 in `verl` using the Megatron-LM backend so that we can efficiently obtain gradient norm statistics. As can be seen in Figure 3, our method reduces the overhead of the naive method from 100% to a 19% step time overhead while only modestly increase memory usage by 14%. See Appendix C for more implementation details.

### 3.5. VCPO: Variance Controlled Policy Optimization

As noted in Appendix E.2, we find that *sequence-level* truncated importance sampling (TIS) with a higher threshold ($c = 8.0$) delays collapse the longest among all current stabilization methods. We thus combine sequence-level TIS with our effective-sample-size step-size scaling and our off-policy optimal baseline. This leads to the following surrogate loss, whose gradient equals our baseline-aware,

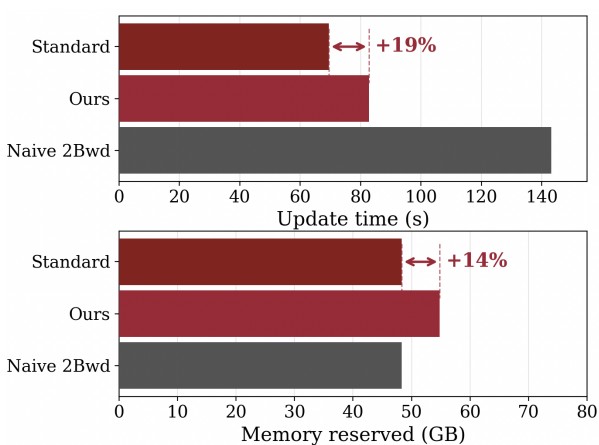

*Figure 3.* Compute (left) and memory (right) overhead of baseline-aware updates for Qwen2.5-7B on 4×H100 GPUs (TP=4) with a sequence length of 8192 tokens.

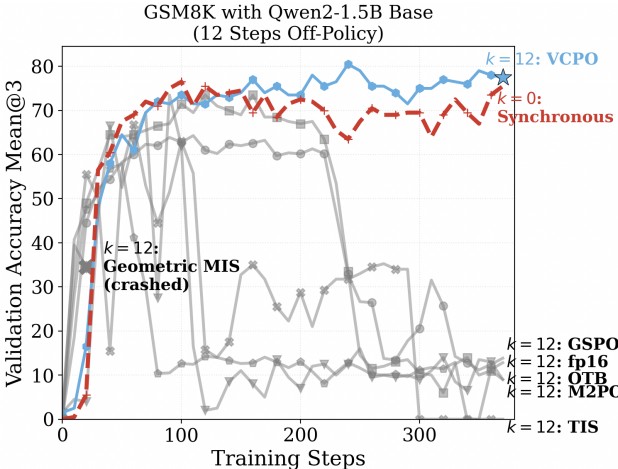

*Figure 4.* GSM8K with Qwen2-1.5B under PIPELINERL-12 (high policy lag). Most baselines lead to training collapse (or crash, e.g. Geometric MIS masks all sequences and has no loss), while VCPO remains stable throughout training and matches synchronous performance. Training details and hyperparameters can be found in Appendix F.2.

truncated IS policy-gradient estimator

$$\mathcal{L}_{\text{VCPO}}(\theta) = -\mathbb{E}_{x\sim\mathcal{D},\, y\sim\mu(\cdot|x)}$$
$$\left[ w^{\text{TIS}}(x,y)\left(R(x,y) - b^{\star}_{\text{OPOB}}\right)\log\pi_\theta(y\mid x)\right]$$
$$\approx -\frac{1}{B}\sum_{i=1}^{B} w_i^{\text{TIS}}\left(R_i - b^{\star}_{\text{OPOB}}\right)\log\pi_\theta(y_i\mid x_i),$$

$$(9)$$

where the sequence-level truncated importance weight is

$$w^{\text{TIS}}(x,y) = \min\left(\text{sg}\left[\frac{\pi_\theta(y\mid x)}{\mu(y\mid x)}\right],\ c\right)$$

We then apply AdamW using the gradient $\nabla_\theta\mathcal{L}_{\text{VCPO}}(\theta)$, but scale learning rate $\eta$ with an ESS-scaled step size

$$\eta_{\text{eff}} = \eta\cdot\sqrt{\frac{\rho_{\text{ess}}^{\text{off}}}{\rho_{\text{ess}}^{\text{on}}}}$$

such each training step size scales with ESS accordingly. Note although we adopt TIS, we still use the *unclipped IS ratios* to calculate ESS.

## 4. Empirical Evaluation

### 4.1. Tasks

We evaluate VCPO across three representative RL post-training pipelines for LLM reasoning and agentic tool use in highly asynchronous regimes. Across all three domains, VCPO remains robust to substantial learner–sampler mismatch, improving throughput while preserving strong learning performance

1. **Mathematical problem solving.** We evaluate on GSM8K (Cobbe et al., 2021), a benchmark of grade-school math word problems, and on MATH (Hendrycks et al., 2021), a dataset of competition-style mathematics problems spanning multiple subjects and difficulty levels. For both benchmarks, we use a binary reward based on exact match of the final numeric answer and train on the official training split while evaluating on the official validation split.

2. **General reasoning (verifiable).** We construct a Countdown-style arithmetic reasoning dataset using Reasoning Gym (Stojanovski et al., 2025), where rewards are verifiable by deterministic checking of the final answer. We sample 9,000 problems for training and hold out 1,000 problems for validation, following prior setups for verifiable-reward reasoning tasks (Liu et al., 2025b).

3. **Long horizon tool-integrated reasoning.** We evaluate multi-turn tool use in the SimpleTIR setting (Xue et al., 2025), where the model must interleave reasoning with external tool calls. We train using the DAPO dataset (Yu et al., 2025) and evaluate on a held-out exam-style benchmark (AIME2025). This setting requires long rollouts (reasoning and multi-turn interactions with tool calls), so training is often dominated by rollout generation. As a result, strictly on-policy training underutilizes compute: the gradient updates must wait for sampling to complete, and this bottleneck worsens at scale and in sparse-reward domains that require more exploration.

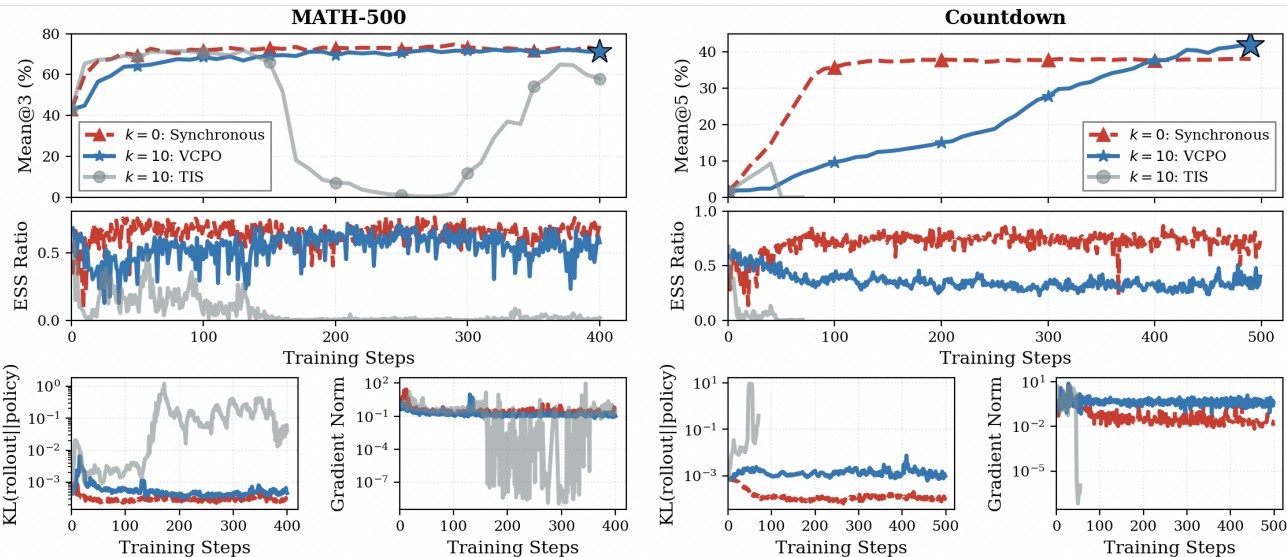

*Figure 5.* Qwen2.5-7B under PIPELINERL-10 (10 Steps Off-Policy) on Countdown and MATH-500. Across both tasks, sequence level truncated importance sampling (TIS) suffers ESS collapse followed by KL/gradient instability and degraded accuracy, whereas VCPO maintains healthy ESS and stable updates, reaching synchronous performance. Training details and hyperparameters are provided in Appendix F.3.

## 4.2. Metrics and Baselines

**On-Policy Training as Oracle Baseline** To disentangle task-learning dynamics from instabilities introduced by asynchrony, we report a fully synchronous ($k = 0$) RL run as an oracle reference (matched training steps and hyperparameters).

**Other Baselines.** We compare against representative prior approaches spanning (i) masking/clipping-based stabilization (e.g., M2PO, MIS, TIS), (ii) objective-level modifications such as GSPO and OTB, and (iii) numerical/systems mitigations (e.g. using fp16 precision). Although (ii) and (iii) were not designed specifically for highly asynchronous RL, we include them for completeness because they are commonly used to stabilize on-policy LLM-RL training. In this paper, we train dense models, and so MoE specific methods such as Rollout Routing Replay are not considered.

## 4.3. VCPO Delivers Synchronous Performance with Higher Throughput

Across mathematical reasoning, general reasoning, and tool-use tasks, VCPO matches the final performance of fully synchronous RL even under high asynchrony. On GSM8K, Figures 4 show that VCPO enables stable training while several baselines collapse or crash in this regime. For the Countdown and MATH-500 tasks, Figure 5 shows that VCPO attains the same peak performance as synchronous RL with higher end-to-end throughput enabled by asynchronous training. Table 1 summarizes the end-to-end wall-clock gains of up to 2.5×. Furthermore, component ablations

*Table 1.* End-to-end training times and validation accuracy for synchronous vs. asynchronous training (lag $k$).

| Method | Countdown ↑ | Steps | GPU hours ↓ |
|---|---|---|---|
| Base | 1.6% | – | – |
| Sync ($k$=0) | 38.4% | 400 | 143.2 |
| VCPO + Async ($k$=10) | 41.9% | 400 | **89.6** |

| Method | MATH-500 ↑ | Steps | GPU hours ↓ |
|---|---|---|---|
| Base | 40.2% | – | – |
| Sync ($k$=0) | 72.0% | 400 | 134.4 |
| VCPO + Async ($k$=10) | 71.6% | 400 | **92.8** |

| Method | AIME 2025 ↑ | Steps | GPU hours ↓ |
|---|---|---|---|
| Base | 5.3% | – | – |
| Sync ($k$=0) | 26.7% | 300 | 420.2 |
| VCPO + Async ($k$=2) | 27.8% | 220 | **168.9** |

show both ESS based learning rate scaling and the off-policy optimal baseline improve robustness, and combining them yields the best stability and accuracy (Appendix D).

## 4.4. VCPO Provides State-of-the-Art Stability Under High Asynchrony

**Enabling Longer Stable Training.** Figure 4 summarizes our highly asynchronous training results (12 steps off-policy) on GSM8K with Qwen2-1.5B. While the synchronous baseline trains smoothly, most existing stabilization approaches fail under this level of policy lag.

In contrast, VCPO consistently avoids collapse. Across baselines: (i) ratio-based clipping (TIS) and masking (MIS)

eventually collapse; (ii) algorithmic variants such as GSPO and OTB do not reliably prevent collapse; and (iii) systems-side mitigations (e.g., FP16) delay collapse but do not address the underlying instability mechanism.

See Appendix E for additional comparisons and ablations, including systematic sweeps over TIS/MIS hyperparameters, algorithmic and systems variants (GSPO, OTB, learning-rate tuning), and tasks.

**Avoiding Effective Sample Size Collapse and Training Degradation.** As shown in Figure 5, in both general mathematical (MATH-500) and reasoning tasks (Countdown), naively training under high asynchrony leads to ESS-ratio collapse followed by gradient instability and performance degradation. However, VCPO remains stable, maintaining a high ESS ratio, stable gradient norms, and KL levels throughout training.

**Scaling to extreme staleness.** While the end-to-end training speedups from asynchronous are fully saturated for $k = 10$ steps off-policy, for completeness we stress test stability as policy lag increases. In Figure 6, we see VCPO training remains stable up to at least $k=128$ steps off-policy while matching synchronous training performance ($k=0$).

### 4.5. VCPO Enables Fast and Stable Long-Horizon RL

We further evaluate VCPO in a long-context, tool-integrated multi-turn setting following SimpleTIR (Xue et al., 2025), where rollouts are substantially longer and training is known to be more brittle (Li et al., 2026). We train with minibatch size 128 and 16 responses per prompt, using a maximum completion length of 12K tokens and up to 5 tool calls per trajectory. We train on the DAPO (Yu et al., 2025) dataset and evaluate on AIME 2025, see Appendix F.4 for more details.

Due to longer trajectories and prevalence of out-of-distribution tool calls, we use a lower level of asynchrony (2 steps off-policy). Figure 1 shows that unlike sequence-level TIS which collapses, VCPO improves steadily in both train reward and AIME2025 accuracy, reducing end-to-end training time–a by $2.5\times$ compared the synchronous baseline. This demonstrates that our variance-control mechanisms transfer to long-horizon, multi-turn post-training.

### 5. Discussion

In this work, we introduce VCPO, a drop-in method for robust, off-policy asynchronous RL LLM post-training. We identify a consistent failure mode in highly asynchronous regimes– an effective sample size collapse that amplifies policy gradient variance–and building on this diagnosis, VCPO stabilizes training with two main contributions (i) *effective-sample-size–guided step-size rescaling* to auto-

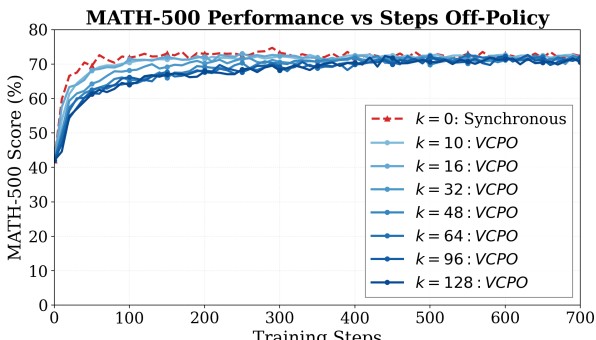

*Figure 6.* **Stable training under extreme policy lag**: Qwen2.5-7B trained with PipelineRL at policy lag $k$ (steps off-policy). VCPO is stable up to at least $k=128$ steps off-policy and matches the synchronous baseline ($k=0$).

matically damp unreliable updates, and (ii) a *closed-form minimum-variance baseline* for importance-weighted gradients that avoids training an auxiliary value model and adds minimal overhead. Across mathematical reasoning, general reasoning, and long-context multi-turn tool use, VCPO improves robustness under substantial learner–sampler mismatch while preserving the throughput benefits of asynchronous rollouts, highlighting explicit variance control as a key ingredient for reliable asynchronous RL at scale.

**Limitations and future work.** We study dense Transformer policies under standard precision and a fixed set of asynchronous pipeline designs. Future work should evaluate VCPO on MoE models (where routing can amplify mismatch), under more aggressive quantization (e.g., FP8), and in substantially longer-horizon agentic settings (e.g., search/planning with sparse rewards), where additional exploration and memory mechanisms may be needed alongside variance control.

## Acknowledgements

We thank MIT-IBM Watson AI Lab, Amazon, Dell, and National Science Foundation for supporting this research. We thank NVIDIA for donating the DGX server.

## Impact Statement

More stable asynchronous RL can reduce wasted compute and enable more efficient post-training for reasoning and tool use. However, improving reliability may also increase agent capabilities and potential misuse; deployments should therefore follow domain-appropriate safety evaluations and monitoring.

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

## A. Why does the Asynchronous RL collapse?

Across model sizes and tasks, we find the same qualitative pattern: runs that remain stable maintain a healthy ESS ratio, while unstable runs exhibit a pronounced ESS-ratio collapse shortly before KL/gradient explosions and performance collapse. This supports our central diagnosis that asynchronous failure is driven by importance-weight degeneracy and dominated off-policy updates.

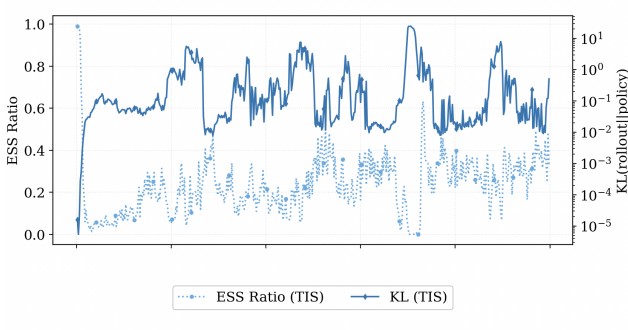

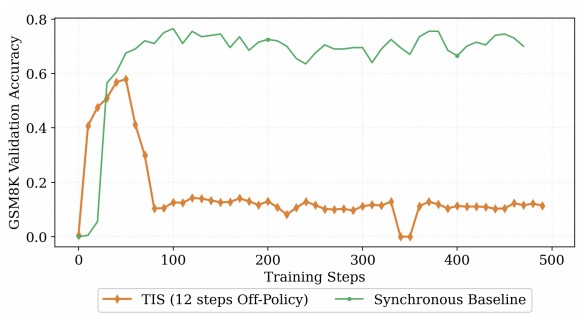

*Figure 7.* Qwen2-1.5B Base on GSM8K questions. Truncated Importance Sampling (TIS) exhibits both effective sample size (ESS) collapse and KL explosion, leading to training collapse.

In addition to Figure 2, which illustrates a collapse trajectory on MATH-500, we observe the same qualitative failure pattern across another mathematical reasoning task (GSM8K) and general reasoning task (Countdown). Figure 7 shows a representative GSM8K run under high policy lag ($k=12$) using truncated importance sampling (TIS): the ESS ratio degrades and then collapses (i.e., the weighted update becomes dominated by a small subset of trajectories), followed by instability in rollout–policy KL and a sharp drop in task performance. Once ESS collapses, updates become effectively single-sample, making the policy-gradient estimator extremely high variance and precipitating abrupt KL spikes. Similarly, on Countdown (Figure 8), validation accuracy collapses to near zero within the first ∼50 training steps, consistent with rapid weight degeneracy and dominated off-policy updates.

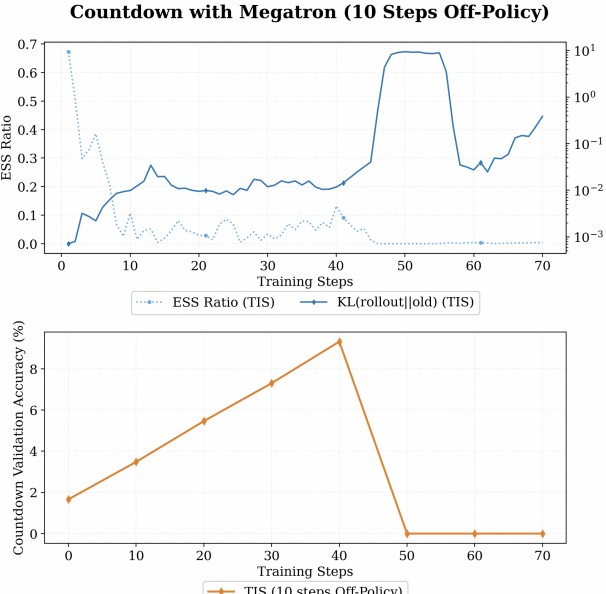

*Figure 8.* Countdown (Qwen2-1.5B Base) under high asynchrony with TIS. Validation accuracy collapses to near zero within the first ~50 training steps, indicating rapid training failure.

## B. Off-Policy Optimal Baseline

We include the derivation of the variance-minimizing *scalar* reward baseline for completeness and to show how the classic importance-weighted optimal-baseline result (Zhao et al., 2013) maps to our asynchronous trajectory-level estimator. Throughout, we assume standard regularity conditions for the score-function gradient estimator, including (i) support: $\mu(\tau) > 0$ whenever $\pi_\theta(\tau) > 0$ so importance ratios are well-defined, and (ii) interchange of gradient and expectation so that $\mathbb{E}_{\tau \sim \pi_\theta}[\nabla_\theta \log \pi_\theta(\tau)] = 0$. These conditions are standard in policy-gradient analyses (Williams, 1992; Sutton et al., 2000; Degris et al., 2012).

### B.1. Single-sample derivation

Let a single trajectory sample $\tau$ have reward (or advantage) $R \in \mathbb{R}$ and score-gradient vector

$$g \triangleq \nabla_\theta \log \pi_\theta(\tau).$$

In the off-policy setting, trajectories are sampled from a behavior policy $\mu$, and we use the (unnormalized) importance ratio

$$w \triangleq \frac{\pi_\theta(\tau)}{\mu(\tau)}.$$

Consider the (single-sample) importance-weighted policy-gradient estimator with a constant baseline $b$:

$$G(b) = w(R - b)g. \tag{10}$$

Subtracting a constant baseline does not change the expected gradient because

$$\mathbb{E}_{\tau \sim \mu}[wg] = \mathbb{E}_{\tau \sim \pi_\theta}[g] = \mathbb{E}_{\tau \sim \pi_\theta}[\nabla_\theta \log \pi_\theta(\tau)] = 0.$$

Therefore, minimizing $\mathrm{Var}(G(b))$ over $b$ is equivalent to minimizing the second moment $\mathbb{E}\|G(b)\|_2^2$.

We have

$$\mathbb{E}\|G(b)\|_2^2 = \mathbb{E}\big[w^2(R - b)^2\|g\|_2^2\big]. \tag{11}$$

Differentiating (11) with respect to $b$ yields

$$\frac{\partial}{\partial b}\mathbb{E}\big[w^2(R - b)^2\|g\|_2^2\big] = -2\,\mathbb{E}\big[w^2(R - b)\|g\|_2^2\big].$$

Setting the derivative to zero gives the variance-minimizing baseline

$$b^\star = \frac{\mathbb{E}\big[w^2 R\|g\|_2^2\big]}{\mathbb{E}[w^2\|g\|_2^2]}. \tag{12}$$

### B.2. Minibatch estimator and plug-in form

For a minibatch of $B$ i.i.d. samples $\{(\tau_i, R_i, w_i, g_i)\}_{i=1}^B$ drawn from $\mu$, the minibatch estimator is

$$\hat{G}(b) = \frac{1}{B}\sum_{i=1}^B w_i(R_i - b)g_i.$$

Since averaging i.i.d. samples scales variance by $1/B$, the same $b^\star$ in (12) minimizes $\mathrm{Var}(\hat{G}(b))$.

In practice, we use the minibatch plug-in estimate corresponding to (12):

$$b^\star \approx \frac{\sum_{i=1}^B w_i^2\|g_i\|_2^2 R_i}{\sum_{i=1}^B w_i^2\|g_i\|_2^2} \tag{13}$$

**Remarks.** (i) Eq. (12) is the same importance-weighted optimal-baseline structure derived by Zhao et al. (Zhao et al., 2013). In the on-policy regime ($w \equiv 1$), it reduces to the classic gradient-norm-weighted optimal baseline (Greensmith et al., 2004; Weaver & Tao, 2013).
(ii) Relative to group-mean baselines $b = \frac{1}{B}\sum_i R_i$, Eq. (13) shows that variance-optimal baselining in the off-policy regime depends on *both* the importance ratios and the gradient magnitudes. OTB concurrently identifies the same structure for its token-level baseline under TIS (Li et al., 2026). Our use here is trajectory-level and exact: we compute the full-model per-trajectory gradient norms used by the plug-in estimator.

## C. Efficient Baseline Aware Gradient Computations

We implement our baseline-aware updates in Megatron-LM, leveraging its explicit control over parameter shards and optimizer state under DP×TP×CP parallelism.

**Per-trajectory gradient-norm statistics under deferred DP synchronization.** Computing the OPOB baseline requires per-trajectory gradient norms $\|g_i\|_2$. A naive implementation would perform two backward passes: one to obtain $\|g_i\|_2$ and a second to form the baseline-adjusted policy gradient. Instead, we compute all required quantities in a *single* backward pass by deferring the data-parallel (DP) gradient all-reduce until after local per-trajectory statistics are computed. Concretely, we run backprop with DP communication disabled so that each worker retains its local gradients, compute the needed per-trajectory scalars locally, and then perform a single DP all-reduce on the final combined gradient.

This design preserves the FLOPs of a single backward pass while replacing the naive two-backward implementation with modest additional memory traffic (for an extra gradient buffer) and a small number of additional scalar reductions.

**Two-buffer accumulation for baseline-aware gradients.** For each trajectory $\tau_i$, we form the per-trajectory negative log-likelihood $\mathcal{L}_i = -\log \pi_\theta(\tau_i)$ and backpropagate once to obtain the score-gradient $g_i = \nabla_\theta \log \pi_\theta(\tau_i)$. During this pass, we accumulate $g_i$ into two gradient buffers: (i) a reward-weighted buffer with coefficient $w_i R_i$, and (ii) a score buffer with coefficient $w_i$. After the loop, we compute the OPOB baseline $b^\star_{\text{OPOB}}$ from the locally accumulated per-trajectory statistics (Algorithm 1) and form the final baseline-aware gradient $\hat{G} = \frac{1}{B}(G_R - b^\star_{\text{OPOB}} G_S)$ without any additional forward/backward pass.

## D. Component Ablations of VCPO

We ablate VCPO into its two core variance-control components: ESS-guided step-size scaling and the off-policy optimal baseline. To quantify each component's contribution under the *same* asynchronous regime and model/task, we compare: (i) **TIS** (sequence-level truncated IS only), (ii) **TIS + ESS step-size scaling** (our step-size rule applied to AdamW), (iii) **TIS + OPOB** (our closed-form off-policy baseline), and All runs use identical hyperparameters except for the ablated component.

Figure 9 shows that both components improve stability and performance relative to TIS alone, and are complementary.

## E. Further VCPO Ablation and Comparison Studies

This appendix provides additional ablations in the MATH and GSM8K setting to isolate the contributions of (i) the ESS-guided step-size rule and (ii) the optimal off-policy baseline as well compare with previous methods. Unless otherwise noted, we follow the same training setup as in the main experiments Appendix F.3).

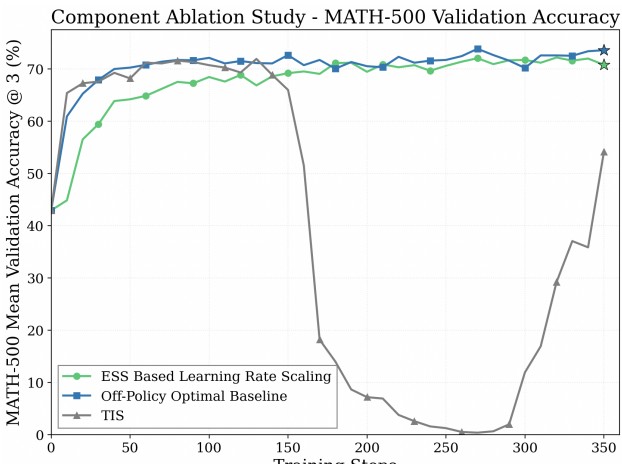

*Figure 9.* Component ablations on MATH-500 under high asynchrony (Qwen2.5-7B, PIPELINERL-10). We compare TIS alone, as well as adding ESS-guided step-size scaling or the off-policy optimal baseline (OPOB). Both components improve stability and accuracy.

### E.1. Is Lowering the Learning Rate Sufficient for Stable Asynchronous RL?

A natural question is whether instability under high asynchrony can be addressed by simply lowering the learning rate. Our ESS-guided rule can be viewed as an *automatic, data-dependent* step-size schedule: it reduces the effective step size precisely when importance weights concentrate, without requiring manual learning-rate sweeps.

To test whether conservative tuning alone is sufficient, we run fixed low-learning-rate baselines under the same asynchronous setting. Figure 10 compares VCPO to a lower learning rate (with lr=$1^{-7}$ instead of lr=$1^{-6}$) variant of truncated importance sampling (TIS). While reducing the learning rate can mitigate catastrophic collapse, it does so by slowing learning and yielding worse final accuracy. In contrast, VCPO achieves both higher accuracy and stable training without per-task learning-rate tuning, illustrating a favorable stability–efficiency tradeoff.

### E.2. Are Clipping or Masking Methods Sufficient?

We next examine whether standard heuristics that directly bound importance ratios are sufficient to stabilize highly asynchronous RL. We report additional results for both *masking* (MIS-style) and *clipping* (TIS-style) applied to different ratio definitions.

**Methods.** We consider three ratio variants for both masking and clipping: (i) **sequence ratio** (full-trajectory likelihood ratio), (ii) **token ratio** (per-token likelihood ratio), and (iii) **geometric mean ratio** (a length-normalized proxy). For masking, we zero out the update when the chosen ratio

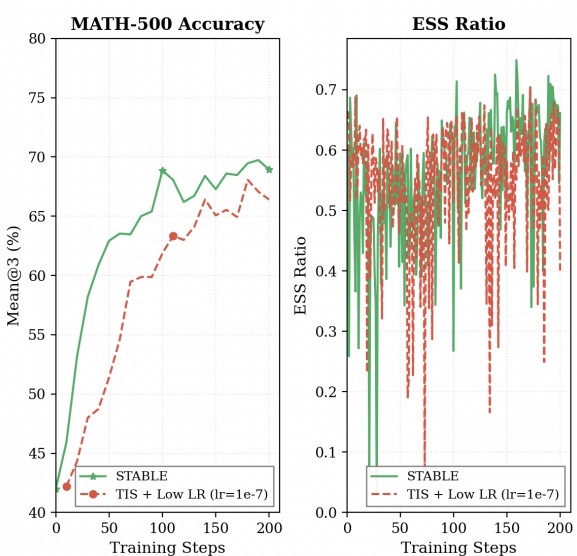

*Figure 10.* Lowering the learning rate is not a substitute for variance-aware step-size control. On MATH-500 under high asynchrony, a very small fixed learning rate (TIS + Low LR) improves stability but converges more slowly and to lower accuracy, despite exhibiting similar ESS levels. VCPO reaches higher accuracy with stable updates without learning-rate sweeps.

exceeds a threshold; for clipping, we cap the ratio at the threshold.

DeepSeek-v3.2 (DeepSeek-AI et al., 2025) introduces a masking heuristic that is closely related to masked importance sampling, using a *geometric-mean* MIS with an *additional sign constraint on advantages* to discard unstable updates.

**Thresholds.** Following prior work and practitioner guidance suggesting that bounding extreme ratios can reduce variance (Yao et al., 2025a; Wu et al., 2025; Fu et al., 2025), we evaluate two representative maximum thresholds, $c \in \{2.0, 8.0\}$, across all ratio variants. (We keep other hyperparameters matched to the main setup.)

In Figures 11 and 12, TIS $(a, b)$ denotes *truncated* importance sampling where ratios outside the interval $(a, b)$ are clipped back into $(a, b)$, while MIS $(a, b)$ denotes *masked* importance sampling where updates with ratios outside $(a, b)$ are discarded (i.e., assigned zero weight).

**Findings and tuning sensitivity.** Figures 11 and 12 show that these masking/clipping methods can delay collapse in some settings, but it is not a reliable substitute for VCPO under high asynchrony. In particular, sequence-level MIS is highly brittle: at both thresholds we find it can mask nearly all sequences, effectively removing the learning signal and causing training to fail. Token-level variants can be less brittle but remain unstable and sensitive to the threshold

choice. Among the baselines we tested, *sequence-level TIS with the larger threshold ($c = 8.0$) performs best*, but still underperforms VCPO and does not consistently prevent late-stage instability.

A further limitation is additional overhead of hyperparameter tuning: the importance-ratio distribution becomes broader and more heavy-tailed as model sizes increases, moving to lower preivios, so the stabilizing threshold depends strongly on the degree of off-policy-ness. In practice this requires additional sweeps over both the threshold and the ratio definition (sequence vs. token vs. geometric mean), which becomes increasingly costly when scaling to new tasks, models, or lag values.

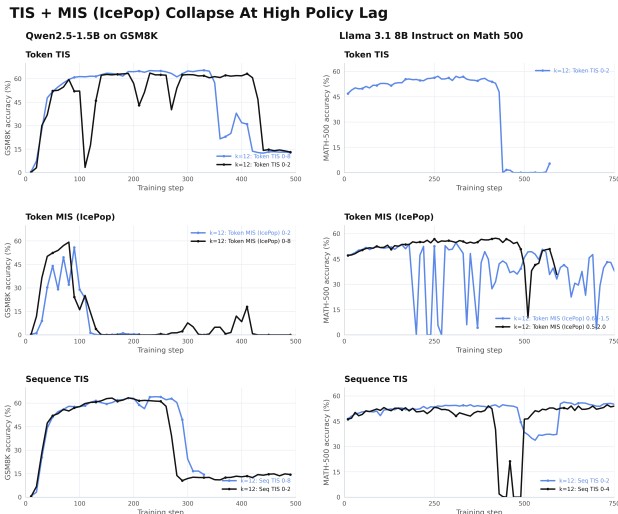

*Figure 11.* **TIS and MIS failures** under high policy lag across Qwen2.5-1.5B on GSM8K and Llama3.1-8B Instruct on MATH-500. Although some thresholds delay collapse, ratio clipping/masking alone does not reliably prevent failure under high asynchrony.

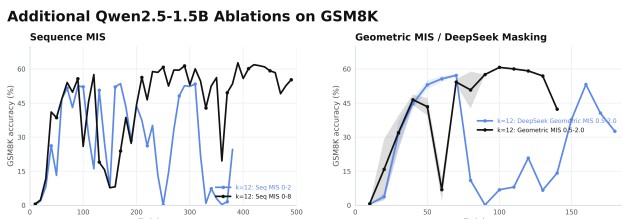

*Figure 12.* **Additional Qwen2.5-1.5B masking ablations** on GSM8K under high policy lag. Sequence-level MIS and geometric-mean MIS / DeepSeek-style masking are sensitive to threshold choices and can become unstable.

**M2PO-style masking.** We reproduce M2PO (Zheng et al., 2025b) using the authors' recommended masking threshold (max $= 0.04$). Figure 13 shows that at high policy lag (PIPELINERL$k{=}12$) on GSM8K with Qwen2-1.5B,

M2PO does not yield stable training: the trusted-token (mask) fraction becomes highly erratic and can collapse toward masking nearly all tokens, coinciding with KL spikes and a sharp drop in validation accuracy. We note reducing the lag to $k=10$ does not reliably fix this behavior. We suspect that the commonly adopted PIPELINERL style asynchronous training setup may differ from the exact regime studied in M2PO. In particular, their pipeline appears to train on exclusively stale rollouts from a fixed-lag behavior policy, whereas our system maintains a queue with *mixed-staleness* samples and may also include in-flight updates with partially completed rollouts. This additional heterogeneity can broaden the importance-ratio distribution and may exacerbate masking instability, so the results in Figure 13 should be interpreted as evaluating M2PO under our (more heterogeneous) PIPELINERL style asynchronous stack.

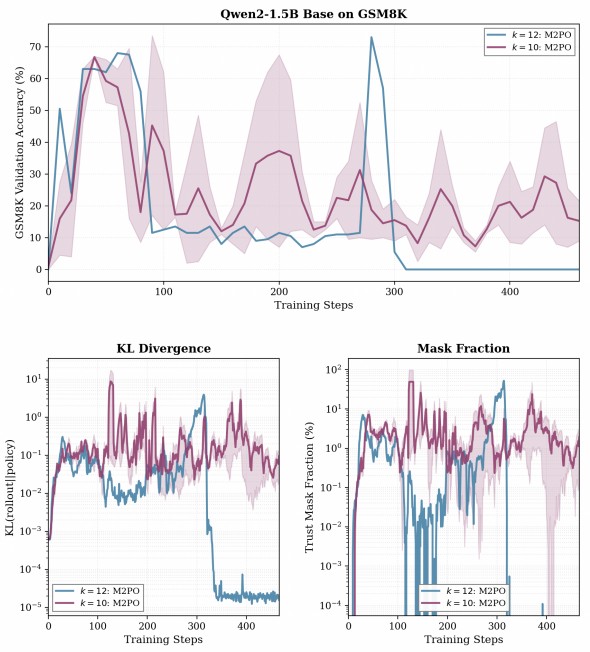

*Figure 13.* M2PO under high asynchrony on GSM8K (Qwen2-1.5B). Top: GSM8K validation accuracy. Bottom-left: rollout–policy KL divergence (log scale). Bottom-right: trusted-token mask fraction (log scale). At $k=12$ (and similarly at $k=10$), the mask fraction becomes unstable and can collapse toward masking nearly all tokens, coinciding with KL instability and training collapse. Shaded regions indicate run-to-run variability.

### E.3. Does KL Loss Improve Stability?

A common stabilization method is to add a KL regularizer (Schulman et al., 2017; Shao et al., 2024) toward a reference policy, either as an explicit penalty in the objective or by folding it into the reward. To test whether KL regularization alone can mitigate the collapse we observe under large

policy lag, we follow prior work (Shah et al., 2026) which recommends a KL penalty to the reward (i.e., no direct gradients through the KL term)

$$R'(x,y) \;=\; R(x,y) \;-\; \beta \, \mathrm{KL}(\pi_\theta(\cdot \mid x) \,\|\, \pi_{\mathrm{ref}}(\cdot \mid x)).$$

We use a coefficient $\beta = 0.001$ and apply this modification on top of the SEQ-TIS baseline in a highly asynchronous setting ($k = 12$). As shown in Figure 14, adding KL-in-reward does not prevent collapse: training still eventually becomes unstable and fails. Moreover, KL regularization reduces peak performance, yielding a lower best validation accuracy than SEQ-TIS without the KL term.

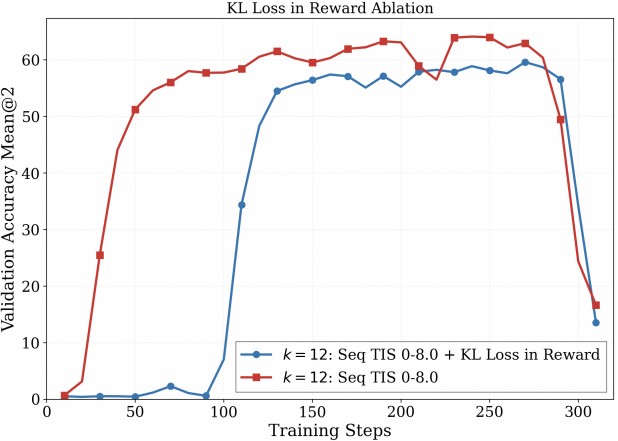

*Figure 14.* Effect of adding a KL penalty in the reward for SEQ-TIS under high asynchrony ($k = 12$) on GSM8K with Qwen2-1.5B. KL-in-reward ($\beta = 0.001$) does not prevent eventual collapse and also lowers the peak validation accuracy compared to SEQ-TIS without KL.

### E.4. Are Gradient-Norm Proxies Like Response Length or Logit "Energy" Sufficient?

Several recent baselines reduce implementation cost by replacing exact gradient norms with cheaper proxies. OPO uses response length as a proxy for gradient magnitude (Hao et al., 2025):

$$\|g_i\| \propto L_i, \tag{OPO}$$

motivated by the intuition that longer responses contribute more log-probability terms.

OTB derives a token-level optimal baseline and introduces a logit-gradient "energy" proxy for the token score-function norm (Li et al., 2026). By defining the cumulative per-token *score-function norm*

$$W_t \triangleq \sum_{j=1}^{t} \|\nabla_\theta \log \pi_\theta(y_j \mid x, y_{<j})\|_2^2, \tag{OTB}$$

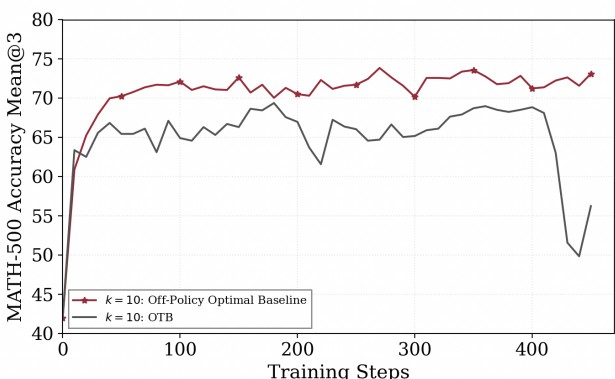

*Figure 15.* Qwen2.5-7B Base trained on MATH-500 under high asynchrony ($k$=10): the logit-energy proxy baseline (OTB) eventually becomes unstable and collapses, while our off-policy optimal baseline (OPOB) remains stable and sustains higher validation accuracy throughout training.

and approximate the per-token score norm using a closed-form weight

$$\hat{w}_t = 1 - 2\,\pi_\theta(y_t \mid x, y_{<t}) + \|\boldsymbol{\pi}_t\|_2^2, \qquad (14)$$

where $\|\boldsymbol{\pi}_t\|_2^2 = \sum_{v \in \mathcal{V}} \pi_\theta^2(v \mid x, y_{<t})$ is the sum of squared next-token probabilities.

OTB's Section 5.3 motivates this proxy analytically for Transformers with RMSNorm final hidden states: the last-layer gradient norm decomposes into a hidden-state factor and a vocabulary-distribution factor, and under an approximate constant hidden-state norm the latter can be computed from logits.

However, the assumptions underlying this proxy have not been verified for full-parameter gradient norms in the asynchronous LLM-RL regimes we study. Empirically, we find that the logit-energy proxy is not a reliable estimate of the exact per-trajectory full-model gradient norms needed by the variance-minimizing baseline. Figure 16 directly compares these proxies with exact full-model gradient norms, and Figure 15 shows that substituting the OTB proxy into our off-policy baseline is less stable than using exact gradient norms. These results suggest that the proxy can miss important full-parameter gradient variation under our models and tasks.

## F. VCPO Training Details

### F.1. Optimization Hyperparameters

We list the hyperparameters for optimization used in all our experiments in Table 2.

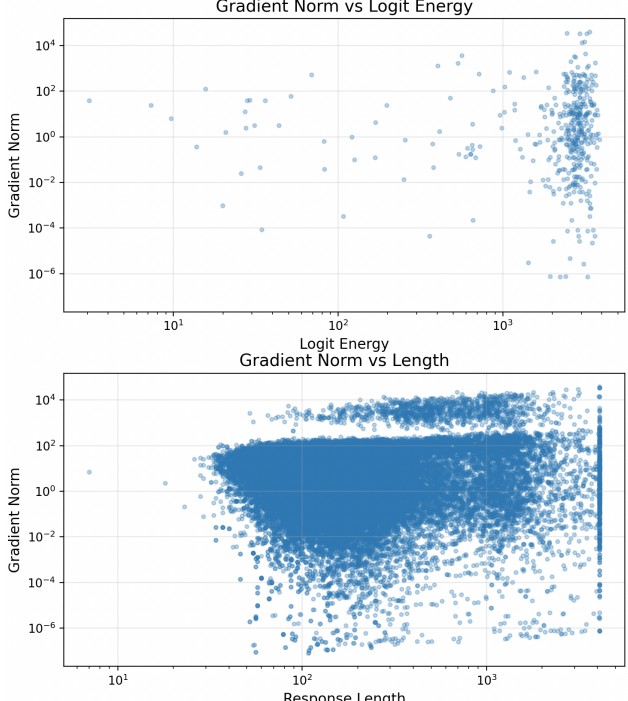

*Figure 16.* Qwen2-1.5B Base trajectories on GSM8K questions, with max response length capped to 4096 tokens

*Table 2.* Optimization hyperparameters

| Hyperparameter | Value |
|---|---|
| Optimizer | AdamW |
| Learning rate | $1 \times 10^{-6}$ |
| Warmup steps | 0 |
| Weight decay | 0.1 |
| AdamW $\beta_1, \beta_2$ | 0.9, 0.999 |
| AdamW $\epsilon$ | $10^{-8}$ |
| Gradient clipping | 1.0 |
| Entropy coefficient | 0.0 |
| KL coefficient | 0.0 |

### F.2. GSM8K

**Task and reward.** We train on the GSM8K training split (Cobbe et al., 2021) using a binary reward that checks whether the model's final numeric answer exactly matches the ground truth. We evaluate on the GSM8K test set and report accuracy (exact-match) over the full set.

**Asynchronous setup.** We use PIPELINERL-$k$ asynchronous rollouts with maximum policy lag $k$=12, and compute sequence-level importance weights between the sampler and learner policies. We use 6× A6000 GPUs for sampling and 2× A6000 GPUs for training. All methods (including our baselines) share the same sampling/training pipeline and differ only in the variance-control and/or masking rules applied to the policy-gradient update.

**Note on Hardware.** Prior work (Liu et al., 2025a) reports that Ampere-generation GPUs can exhibit larger sampler–learner numerical discrepancies (e.g., in bf16 log-prob computations) which are not observed on Hopper-generation hardware. This leads to training collapse even for fully on-policy RL without policy-lag. We intentionally run GSM8K on Ampere (A6000) to stress-test robustness under amplified mismatch in addition to policy lag. Accordingly, this serves as a worst-case stress test: absolute metrics may change on other hardware, but the relative stability gains of our method over baselines are consistent in our checks.

*Table 3.* Training hyperparameters for the GSM8K task

| Hyperparameter | Value | Reference |
|---|---|---|
| Model | Qwen2-1.5B | |
| Learning Rate | $1 \times 10^{-6}$ | |
| Learning Rate Schedule | Warmup Stable Decay | |
| Learning Rate Warmup Steps | 5 | |
| Learning Rate Stable Steps | 395 | |
| Learning Rate Decay Steps | 0 | |
| Generation Temperature | 1.0 | |
| Max Prompt Token Length | 1024 | |
| Response Length | 2048 | |
| Number of Prompts per Batch | 8 | |
| Number of Completions per Prompt | 8 | |
| Batch Size (effective) | 64 | |
| Number of Training Steps | 400 | |
| Total Prompts Seen | 3200 | |
| Total Episodes | 25600 | |
| *Asynchronous-specific hyperparameters* | | |
| Max Staleness | 12 | PIPELINERL-$k$ value |
| Number of Sampler GPUs | 6 | |
| Number of Trainer GPUs | 2 | |
| Max IS ratio | 2.0/8.0 | Max value from TIS |
| $\rho_{ess}^{on}$ | 1.0 | On-policy ESS ratio (See 5) |

## F.3. Countdown and MATH

**Countdown (Reasoning Gym).** We build a Countdown-style arithmetic reasoning dataset using Reasoning Gym (Stojanovski et al., 2025). Rewards are verifiable by deterministic checking of the final answer; we use 9,000 problems for training and 1,000 held-out problems for validation.

**MATH.** We train on the official MATH training split (Hendrycks et al., 2021) with an exact-match final-answer reward. We report validation accuracy on the standard MATH validation split, and additionally use the MATH-500 subset for controlled comparisons in the main paper.

For the Llama3.1-8B Instruct diagnostic in Figure 2, we follow the math-RL setup described by Perplexity (Perplexity AI Team, 2025): the model is initialized from the vanilla Llama3.1-8B Instruct checkpoint, trained on the ORZ math dataset, and evaluated on MATH-500. We train this diagnostic run with sequence-level TIS in our asynchronous setting.

**Asynchronous setup.** For both tasks, we run PIPELINERL-$k$ with maximum policy lag $k=10$. We use $4 \times$H100 GPUs for sampling and $4 \times$H100 GPUs for training.

*Table 4.* Training hyperparameters for the MATH and Countdown tasks. The Llama3.1-8B Instruct column corresponds to the Figure 2 diagnostic run.

| Hyperparameter | Qwen2.5-7B | Llama3.1-8B Instruct |
|---|---|---|
| Model | Qwen2.5-7B Base | Llama3.1-8B Instruct |
| Training data | MATH / Countdown | ORZ math dataset |
| Evaluation | MATH-500 / Countdown val | MATH-500 |
| Learning Rate | $1 \times 10^{-6}$ | $5 \times 10^{-7}$ |
| Learning Rate Schedule | Warmup Stable Decay | Warmup Stable Decay |
| Learning Rate Warmup Steps | 0 | 0 |
| Learning Rate Stable Steps | 400 | 1000 |
| Learning Rate Decay Steps | 0 | 0 |
| Generation Temperature | 1.0 | 1.0 |
| Max Prompt Token Length | 2048 | 2048 |
| Response Length | 2048 | 8192 |
| Number of Prompts per Batch | 32 | 32 |
| Number of Completions per Prompt | 16 | 16 |
| Batch Size (effective) | 512 | 512 |
| Number of Training Steps | 400 | 1000 |
| Total Prompts Seen | 12,800 | 32,000 |
| Total Episodes | 204,800 | 512,000 |
| *Asynchronous-specific hyperparameters* | | |
| Max Staleness | 10 | 12 |
| Number of Sampler GPUs | 4 | 4 |
| Number of Trainer GPUs | 4 | 4 |
| Off-policy correction | Sequence-level TIS | Sequence-level TIS |
| Max IS ratio | 8.0 | 8.0 |
| $\rho_{ess}^{on}$ | 1.0 | 1.0 |

## F.4. Multi-Turn Tool Integrated Reasoning

**Task, data, and evaluation.** We follow the SimpleTIR setting (Xue et al., 2025) for multi-turn tool-integrated reasoning. We train on the DAPO-17K dataset (Yu et al., 2025) and evaluate on held-out exam benchmarks (AIME 2025), reporting exact-match final-answer accuracy. We use Qwen2.5-7B Base (rather than newer Qwen3 variants) to reduce the risk of evaluation-set contamination.

**Rollout Configuration and Asynchrony.** We generate up to 5 turns per episode with a maximum completion length of 12,288 tokens (Table F.4). We run asynchronous training with moderate policy lag (PIPELINERL-2) and compute sequence-level importance weights for off-policy correction.

We use $2\times$ B200 GPUs for sampling and $2\times$ B200 GPUs for training

*Table 5.* Training hyperparameters for the multi-turn tool integrated reasoning

| Hyperparameter | Value | Reference |
|---|---|---|
| Model | Qwen2.5-7B | |
| Learning Rate | $1 \times 10^{-6}$ | |
| Learning Rate Schedule | Warmup Stable Decay | |
| Learning Rate Warmup Steps | 0 | |
| Learning Rate Stable Steps | 200 | |
| Learning Rate Decay Steps | 0 | |
| Generation Temperature | 1.0 | |
| Max Prompt Token Length | 2048 | |
| Response Length | 12,288 | |
| Number of Prompts per Batch | 128 | |
| Number of Completions per Prompt | 16 | |
| Batch Size (effective) | 2048 | |
| Number of Training Steps | 200 | |
| Total Prompts Seen | 25,600 | |
| Total Episodes | 409,600 | |
| *Asynchronous-specific hyperparameters* | | |
| Max Staleness | 2 | PIPELINERL-$k$ value |
| Number of Sampler GPUs | 2 | |
| Number of Trainer GPUs | 2 | |
| Max IS ratio | 8.0 | Max value from TIS |
| $\rho_{\text{ess}}^{\text{on}}$ | 0.55 | On-policy ESS ratio (See 5) |

