# OpenReview forum: "Stable Asynchrony: Variance-Controlled Off-Policy RL for LLMs"
_ICML.cc/2026/Conference — ICML 2026 regular_

### Official Review · Reviewer_JzMG · 2026-02-13

**Soundness:** 3
**Presentation:** 2
**Significance:** 2
**Originality:** 2
**Overall Recommendation:** 4
**Confidence:** 3

**Summary:**

This paper studies RL with verifiable rewards (RLVR) for LLMs under modern disaggregated/asynchronous training pipelines where rollouts are generated by a stale behavior policy $\mu$ while updates are applied to a newer learner policy $\pi_\theta$. The authors argue that in high-lag regimes, importance ratios $w(\tau)=\pi_\theta(\tau)/\mu(\tau)$ become heavy-tailed, the effective sample size $ESS=\frac{(\sum_i w_i)^2}{\sum_i w_i^2}$ collapses, and updates become dominated by a few trajectories, causing spikes in KL and eventual training collapse. To stabilize training, they propose VCPO, which combines (i) sequence-level truncated importance sampling (TIS) and (ii) two variance-control components: an $ESS$-guided step-size rule  and an off-policy variance-minimizing baseline
implemented efficiently via a single backward pass using two gradient buffers. Experiments across reasoning tasks and asynchronous settings report substantially improved stability (avoiding collapse) and competitive compute-normalized performance relative to synchronous baselines and several stabilization baselines.

**Compliance With Llm Reviewing Policy:**

Affirmed.

**Key Questions For Authors:**

see weakness.

**Limitations:**

yes

**Strengths And Weaknesses:**

## Strengths
The variance-control framing is coherent: off-policy correction via $w_i$ is variance-limited in practice, and monitoring $ESS$ is an appropriate diagnostic for weight degeneracy. The proposed step-size rule $\eta_{\mathrm{eff}}=\eta\sqrt{ESS/B}$ is intuitive as an effective batch size'' analog. The closed-form baseline $b^*_{\mathrm{OPOB}}$ is a reasonable extension of classic minimum-variance baselines to importance-weighted gradients, and the single-backward implementation is practically valuable. \item \textbf{Weaknesses:} The central motivation risks being overstated: instability is attributed primarily to off-policy-ness'' from asynchrony, but the paper does not fully disentangle (a) actor--learner staleness, (b) multiple policy updates per rollout (PPO-style epochs), (c) reward/advantage scale changes, and (d) optimizer dynamics. Since these different sources induce different ratio distributions, it is unclear how general the collapse mechanism is across common RLVR recipes. In addition, TIS introduces bias; the paper positions variance as the main issue, but it remains unclear when bias becomes the limiting factor as staleness grows or horizons lengthen.

Stabilizing asynchronous RLVR is practically important: disaggregated pipelines are the dominant way to scale long-horizon rollouts, and training collapse is a severe failure mode. A low-overhead method that improves stability without heavy synchronization or expensive hyperparameter sweeps could have meaningful systems impact.

## weakness
The paper’s motivation is strongest when framed as: in disaggregated asynchronous pipelines with large staleness and long horizons, sequence-level ratios become heavy-tailed, causing $ESS \ll B$ and dominated updates; thus variance control is necessary for stable learning at high throughput.
However, the broader literature suggests ``off-policy'' is not a monolith: some settings described as off-policy (e.g., multiple learner updates per rollout under clipping/constraints) can work well, and some algorithms appear inherently more robust to stale data. The paper would benefit from explicitly contrasting these regimes and clarifying why the targeted regime is uniquely failure-prone.
Many ingredients have close precedents in RL systems (importance truncation, $ESS$ monitoring, minimum-variance baselines, variance control under latency). The incremental nature is fine for ICML/ICLR if the empirical story is strong, but the paper should be more explicit about what is \emph{new versus adapted} and why prior fixes (e.g., gating on $ESS$, alternative weighting schemes) are insufficient.

https://arxiv.org/pdf/2510.11696

---

> ### Author Rebuttal · Authors · 2026-03-31
>
> ### 1. Disentangling Sources of Instability
>
> We agree the four factors listed are all relevant, but for importance-sampled policy-gradient objectives they all interact through the same core mechanism: sampler $\mu$ and learner $\pi_\theta$ policy divergence produces heavy-tailed importance ratios $w(\tau) = \frac{\pi_\theta(τ)}{\mu(\theta)}$ and ESS collapse (Figure 1). Actor-learner staleness (a) is this divergence, created by either PPO-style epochs (b) or PipelineRL-style updates. Optimization dynamics (c) determine how staleness accumulates. Reward/advantage scaling (d) affects gradient magnitudes but not the importance-ratio distribution.
>
> **We emphasize our main experiments disentangle these factors directly**: we fix PPO epochs to 1, keep AdamW settings identical, and compare synchronous ($k=0$) and async PipelineRL-$k$ runs under matched settings.
>
> To address the reviewer’s point that off-policy training is not monolithic, we varied the staleness distribution on MATH-500 ($k=10$): (1) FIFO (original), (2) uniform random sampling, and (3) mixed-recent (sample most recent trajectories with probability p, oldest otherwise).
>
> | Staleness Sampling  Procedure | Peak Val Acc. | Step 50 | Step 100 | Step 200 | Step 400 |
> | --- | --- | --- | --- | --- | --- |
> | FIFO + VCPO | 0.72 | 0.64 | 0.68 | 0.69 | 0.71 |
> | Random + VCPO | 0.74 | 0.68 | 0.70 | 0.71 | 0.73 |
> | Mixed p=50% + VCPO | 0.73 | 0.67 | 0.69 | 0.72 | 0.70 |
> | Mixed p=25% + VCPO | 0.74 | 0.66 | 0.70 | 0.71 | 0.73 |
> | Mixed p=75% + VCPO | 0.73 | 0.68 | 0.70 | 0.71 | 0.71 |
> | Synchronous  | **0.75** | 0.69 | 0.72 | 0.73 | 0.71 |
>
> VCPO remains stable and achieves near-synchronous performance across all settings without hyperparameter retuning, suggesting stability is robust across several kinds of “off-policyness”.
>
>
> ### 2. Is Bias or Variance the Limiting Factor in Asynchronous Long-Horizon RL?
>
> Empirically, variance is the limiting factor before bias becomes the bottleneck.
>
> This can be tested directly. The unbiased off-policy estimator uses the sequence-level importance ratio $\frac{\pi_{\theta}(y|x)}{\mu(y|x)}$. Clipping/masking methods (TIS/MIS/M2PO, etc.) modify importance weights to reduce variance at the cost of bias. If bias were the main issue, then removing clipping should help. Instead, we find the opposite: the unbiased estimator collapses quickly, indicating that variance is the dominant failure mode.
>
> |  | Final MATH-500 Acc. ($k=10$) | Final GSM8K Acc. ($k=12$) |
> | --- | --- | --- |
> | Unbiased | 0% | 0% |
> | Unbiased + VCPO | 72% | 78% |
>
> Importantly, both ESS scaling and OPOB introduce no additional bias, suggesting that VCPO allows for **stable, low-variance unbiased training** under high asynchrony.
>
> ### 3. Clarifying Novelty and Relation to Prior Work
>
> Our contribution is distinct from prior work in three ways.
>
> **(1) New diagnosis and ESS-guided step scaling in Asynchronous LLM RL.**
>
> ESS itself is a concept from statistics, and some prior off-policy LLM-RL works have *reported* ESS (e.g. PipelineRL reports token-level ESS statistics).
>
> However, to the best of our knowledge, prior works do not identify **sequence-level ESS collapse as a leading indicator to collapse** nor use that diagnosis to design an optimizer rule. In our paper, we show that under stale rollouts, heavy-tailed **sequence-level** importance ratios drive $ESS \ll B$, which leads to dominated updates, unstable gradient norms, KL spikes, and reward collapse (Figure 1). This diagnosis directly motivates our ESS-guided scaling rule ($\eta_{eff} = \eta \sqrt{ESS/B}$). Furthermore, we are unaware of any prior or concurrent work utilizing "gating on ESS."
>
> **(2) Off-Policy Optimal Baseline, rather than on-policy proxies, together with an Exact Efficient Implementation.**
>
> While minimum-variance baselines are not new in classical RL, we derive it for the **sequence-level off-policy policy gradient** relevant for asynchronous LLM RL, not just the on-policy case. We also introduce an exact single-backward implementation that makes it **practical at scale** for LLM RL(19% step-time overhead; Figure 2). Recent works propose on-policy length-based (OPO [2025]) or logit-based (OTB [2026] approximations, but our experiments show these proxies correlate poorly with true gradient norms and have suboptimal performance (Appendix E.4).
>
> **(3) Why Prior Methods are Insufficient**
>
> The most relevant prior methods are weighting/clipping/masking (e.g., token/sequence-level IS variants, GSPO, MIS, TIS, M2PO-style masking) methods that reduce variance by modifying or discarding outlier IS ratio samples, introducing a variance-bias tradeoff. As detailed in Appendix E.2, E.3 and Figure 3, we show these **prior methods are insufficient to prevent training collapse**. On the other hand, VCPO directly reduces and controls gradient variance via the off-policy optimal baseline and ESS-guided step scaling without further introducing bias

---

> > ### Author Rebuttal · Reviewer_JzMG · 2026-03-31
> >
> > The concerns has been addressed. I will raise my score. (Interestingly, the reviewers of my paper also asked about the ESS in my RLVR training, so, it seems the ESS is an important factor in RLVR pipeline)

---

### Official Review · Reviewer_Niud · 2026-03-05

**Soundness:** 3
**Presentation:** 3
**Significance:** 3
**Originality:** 3
**Overall Recommendation:** 5
**Confidence:** 3

**Summary:**

This paper proposes VCPO, a method designed to stabilize reinforcement learning (RL) training under conditions of high asynchrony. VCPO introduces twotechniques. First, it rescales the learning rate based on the effective sample size, which is derived from sequence-level importance sampling ratios. Second, it employs a baseline estimation method aimed at minimizing gradient variance, along with a simple strategy that avoids the need for a double backward pass during baseline estimation. Experiments conducted on math, countdown, and tool-usage tasks demonstrate the effectiveness of the proposed approach.

**Compliance With Llm Reviewing Policy:**

Affirmed.

**Final Justification:**

Overall it is a good paper and I recommend accept.

**Key Questions For Authors:**

N/A

**Strengths And Weaknesses:**

**Strengths**:
1. The two techniques proposed in the paper are principled and well motivated. In particular, scaling the learning rate according to the effective sample size is novel in this domain. While the closed-form optimal baseline itself is not new, the technique used to avoid a double backward pass is insightful and practically useful.
2. The experiments are extensive and cover three different domains.
3. The problem addressed is practical and important.

**Weaknesses**:
1. The ablation study is insufficient. Although two techniques are proposed, their individual and combined effects are not clearly analyzed in the main text. While an ablation study is provided in the appendix (Figure 8), it appears that either scaling the learning rate or using the optimal baseline alone is sufficient to stabilize RL training. The paper does not clearly demonstrate that combining both techniques leads to additional benefits.

---

> ### Author Rebuttal · Authors · 2026-03-31
>
> We thank the reviewer for the encouraging assessment and for recognizing the practical importance of the problem, the principled design of VCPO’s two components, and the breadth of the evaluation. We respond below to the remaining question about their respective roles.
>
> ### 1. What is comparative benefit of the ESS based scaling law and Off-Policy Optimal Baseline?
>
> Both components are practical to use together, and they address variance at **different levels:**
> 1. **ESS-Guided Step-Size Scaling acts as a global control:** When importance weights become wildly heavy-tailed, it uniformly dampens the entire learning step, acting as the global defense against catastrophic KL spikes and algorithmic collapse.
> 2. **The Off-Policy Optimal Baseline ($b^*_{OPOB}$) enhances the gradient signal:** It optimally centers the advantages based on both the importance weight and the gradient magnitude of each specific trajectory, reducing the noise in the gradient estimate.
>
> To explicitly demonstrate how their roles differ, we conducted isolated ablations on the long-horizon, tool-use AIME2025 task (at $k=6$):
>
> | **Method Component(s)** | **Peak Val. Acc.** | **Final Val. Acc.** | **Final KL Divergence** |
> | --- | --- | --- | --- |
> | TIS | 20% | 0% (Collapse @ Step 120) | 0.38 |
> | TIS + OPOB Only | 27% | 22% | 0.0052 |
> | TIS + ESS Scaling | 24% | 21% | **0.0042** |
> | TIS + VCPO (Both) | **28%** | **24%** | 0.0047 |
>
> Each component alone substantially improves stability relative to TIS, but combining both gives the best validation accuracy while maintaining similarly low KL. This supports our intended design: OPOB improves the quality of each update, while ESS scaling protects against unreliable low-ESS steps, and using both together gives the strongest overall performance.

---

> > ### Author Rebuttal · Reviewer_Niud · 2026-04-01
> >
> > Thanks for the rebuttal. My concern is addressed. I will maintain my score (5), which is positive.

---

### Official Review · Reviewer_vYM7 · 2026-03-13

**Soundness:** 3
**Presentation:** 3
**Significance:** 3
**Originality:** 2
**Overall Recommendation:** 4
**Confidence:** 4

**Summary:**

This paper identifies Effective Sample Size (ESS) collapse as the root cause of instability in asynchronous RL for LLMs, and proposes **Variance Controlled Policy Optimization (VCPO)** to address it. VCPO combines two mechanisms: (1) ESS-guided step-size scaling that dampens the learning rate by $\sqrt{ESS/B}$ when importance weights become heavy-tailed, and (2) an Off-Policy Optimal Baseline (OPOB)—a closed-form, minimum-variance scalar baseline for importance-weighted policy gradients. Experiments on reasoning benchmarks show that VCPO remains stable at asynchrony factors up to $k=12$ where all baselines collapse, achieving 1.5–2× wall-clock speedups over synchronous GRPO without accuracy loss.

**Compliance With Llm Reviewing Policy:**

Affirmed.

**Final Justification:**

I'd like to maintain a positive score after the overall rebuttal progress.

**Key Questions For Authors:**

1. **Effective learning rate at $k=12$.** What is the typical value of $\sqrt{ESS/B}$ during training at high asynchrony? If it is consistently very small, VCPO's stability may be trivially achieved by barely updating. A plot of the effective LR multiplier over training would clarify this.

2.  **Variable staleness.** Have you tested VCPO with heterogeneous staleness within a batch? Batch-level ESS scaling may not adequately handle the mixed-staleness scenarios common in production systems.

**Limitations:**

yes

**Strengths And Weaknesses:**

Strengths.

1. The paper goes beyond merely observing that "async RL is unstable" and identifies a concrete, measurable precursor—ESS collapse—that precedes KL divergence spikes and reward drops (Figure 1). This diagnostic is independently valuable and provides a principled foundation for the proposed fix, in contrast to ad hoc clipping or masking heuristics.


2. The proposed two techniques are reasonably motivated and well executed. The baselines are comprehensive.

3. The paper is well-organized with clear writing.

Weaknesses.

1. All benchmarks use binary/rule-based rewards on reasoning tasks. The absence of RLHF tasks with learned reward models—where reward noise interacts with importance-weight variance—leaves the generalizability of VCPO unclear. Binary rewards and sparse feedback may be particularly well-suited to variance reduction, inflating the apparent benefit.

2. Using a fixed $k$ for all samples is a controlled experimental choice, but production pipelines exhibit heterogeneous and time-varying staleness. The batch-level ESS scaling may not handle bimodal staleness distributions (e.g., some samples at $k=2$, others at $k=10$) gracefully, and this is not tested.

---

> ### Author Rebuttal · Authors · 2026-03-31
>
> ### 1. What Is the Typical Effective Learning Rate During Training at High Asynchrony?
>
> For MATH-500 and Countdown tasks ($k=10$), Figure 4 shows VCPO converges to a steady learning rate scaling ratio ($\sqrt{ESS/B} = \eta_{ess} / \eta_{base}$) rate of $\approx0.75$ and $\approx0.63$ respectively, suggesting updates are not trivially small.
>
> For GSM8K ($k=12$) and  TIR on AIME2025 ($k=6$), we present aggregate statistics of learning rate scaling ratio ($η_{eff} / η_{base}$) throughout training
>
> |  | mean | max | min | std |
> | --- | --- | --- | --- | --- |
> | GSM8k | 0.59 | 0.89 | 0.13 | 0.15 |
> | AIME2025 | 0.97 | 1.00 | 0.82 | 0.03 |
>
> The scaling ratio stays in a practical range (mean of 0.59 and 0.97)  and does not collapse to near zero, showing that VCPO is not “stabilizing by barely updating.” Rather, it selectively damps high-variance steps while preserving meaningful learning. We will include these plots and statistics in the camera-ready version for transparency.
>
> ### 2. Scope of the current evaluation: Long-Horizon Sparse/Verifiable Rewards
>
> We agree that learned-reward RLHF is an important setting. Our current evaluation focuses instead on **long-horizon reasoning and tool-use tasks with sparse or verifiable rewards**, because these are among the settings where asynchronous RL is especially compelling in practice: rollouts are long, throughput bottlenecks are severe, and stale-rollout variance becomes a central optimization issue.
>
> ### 3. How does VCPO handle the mixed-staleness scenarios common in production systems?
>
> We ablate three heterogeneous sampling strategies on MATH-500 ($k=10$): (1)  FIFO (original), (2) uniform random sampling from queue, and (3) mixed-recent (sample most recent trajectories with probability p, oldest otherwise).
>
> | Staleness Sampling  Procedure | Peak Val Acc. | Step 50 | Step 100 | Step 200 | Step 400 |
> | --- | --- | --- | --- | --- | --- |
> | FIFO + VCPO | 0.72 | 0.64 | 0.68 | 0.69 | 0.71 |
> | Random + VCPO | 0.74 | 0.68 | 0.70 | 0.71 | 0.73 |
> | Mixed p=50% + VCPO | 0.73 | 0.67 | 0.69 | 0.72 | 0.70 |
> | Mixed p=25% + VCPO | 0.74 | 0.66 | 0.70 | 0.71 | 0.73 |
> | Mixed p=75% + VCPO | 0.73 | 0.68 | 0.70 | 0.71 | 0.71 |
> | Synchronous  | **0.75** | 0.69 | 0.72 | 0.73 | 0.71 |
>
> VCPO remains stable and achieves near-identical performance. Theoretically, batch-level ESS scaling naturally adapts because it depends only on the realized importance weights in the current minibatch, not on the underlying staleness distribution. No additional hyperparameter changes were needed.

---

> > ### Author Rebuttal · Reviewer_vYM7 · 2026-04-07
> >
> > Thanks for your responses. My questions are all resolved and I'd like to meaintain my positive score.

---

### Official Review · Reviewer_pSXS · 2026-03-15

**Soundness:** 3
**Presentation:** 3
**Significance:** 3
**Originality:** 3
**Overall Recommendation:** 4
**Confidence:** 4

**Summary:**

The paper proposes an off-policy RLVR method to promote Stable Asynchrony with two key components: an adaptive learning rate based on ESS computation and an optimal baseline derivation for policy gradients.

**Compliance With Llm Reviewing Policy:**

Affirmed.

**Final Justification:**

My concerns are well addressed, and I will increase my score.

**Key Questions For Authors:**

- What is the computational cost of computing the optimal baseline for policy gradients, and any empirical report on the running time?

- The experiments only demonstrate the performance gain to TIS. There is an increasing number of off-policy RLVR algorithms. Can the proposed method also be deployed to other off-policy algorithms, and in what ways can it help general off-policy algorithms?

- The method contains an adaptive learning rate based on ESS computation and an optimal baseline derivation for policy gradients. Which one is more essential? (I guess maybe the adaptive learning rate. In RL, the optimal baseline derivation induces much overhead, and a simple baseline of the mean return is usually applied.)

**Limitations:**

Yes.

**Strengths And Weaknesses:**

Pros:
- The Stable Asynchrony in off-policy RL is an important challenge in research and engineering implementation.
- The proposed adaptive learning rate and optimal baseline derivation are well principled.
- The Baseline-Aware Gradient Computation is natural and saves computation.

Cons:
- Lack of adequate experiments. This is the main weakness in the current version, in my opinion. The paper mainly demonstrates VCPO's performance compared to collapsed TIS. Any algorithm can collapse during training without careful tuning, especially for RL algorithms. The original TIS paper also demonstrates stable learning performance. The comparison should be fair enough.
- Lack of ablation studies to assess the respective contribution of the adaptive learning rate and the optimal baseline derivation. Lack of more insightful analysis to delve into the off-policy training process.
- Lack of discussion or comparison to more off-policy RLVR algorithms.

---

> ### Author Rebuttal · Authors · 2026-03-31
>
> ### 1. Fairness and Adequacy of the Experimental Comparisons
>
> Sequence-level TIS was selected as our primary baseline **because it was the strongest masking/clipping baseline.** In Appendix E.2 (Figures 10, 11) and Figure 3, we compared a broad family of alternatives, including sequence/token-level variants of truncation/masking (TIS/MIS), geometric MIS, DeepSeek-style masking, and M2PO, each across two threshold settings. Among these, sequence-level TIS with a larger clipping threshold was the *most competitive baseline*.
>
> To further address fairness, we report an extended TIS hyperparameter sweep on MATH-500 ($k=10$), but across this sweep TIS still collapses:
>
> | Method | Step 100  | Step 250 | Step 360 | Step 380 | Status |
> | --- | --- | --- | --- | --- | --- |
> | Seq-TIS 8.0 | 0.70 | **0.0** | 0.0 | 0.0 | Collapse @ Step 150  |
> | Seq-TIS 4.0 | 0.69 | **0.0** | 0.0 | 0.0 | Collapse @ Step 160  |
> | Seq-TIS 2.0 | 0.70 | 0.71 | 0.66 | **0.35** | Collapse @ Step 350 |
>
> Note this does not contradict prior TIS results, as the original TIS paper (Yao et al., 2025) studied small train-inference mismatches (quantization/kernel) for the **near on-policy regime**, whereas our paper focuses on the much harder regime of highly asynchronous training.
>
> ### 2.  Compatibility of VCPO with other policy-gradient methods
>
> VCPO introduces two variance-targeted controls for any **importance-weighted score-function policy gradients**, not a new RL objective. Thus it is compatible with critic-free RLVR objectives such as REINFORCE, GRPO, RLOO, and GSPO/GMPO-style updates whenever the gradient can be written as a weighted score-function estimator. **ESS-guided step scaling** applies whenever the method uses (or modifies) importance sampling non-uniform importance weights. **OPOB** is even more general: it applies to any weighted score-function gradient, including the on-policy case (where the importance weights are uniform).
>
> | Component | Applicable scope |
> | --- | --- |
> | ESS-guided LR scaling | Any importance-weighted PG: off-policy REINFORCE, GRPO , GSPO/GMPO, RLOO |
> | OPOB | Any weighted score-function PG (on-policy or off-policy): REINFORCE, GRPO, GSPO/GMPO, RLOO |
>
> ### 3. What is the computational cost of computing the optimal baseline for policy gradients, and any empirical report on the running time?
>
> We report the computational overhead of OPOB in Figure 2. Our single-backward method adds only **19% update-time overhead** relative to a standard backward pass, avoiding a naïve double-backward implementation. More importantly, this modest overhead is outweighed by the global benefit of stable high-asynchrony training: VCPO yields **1.45× / 1.6× / 2.5×** end-to-end speedups on MATH-500 / Countdown / AIME2025, respectively, while matching the synchronous baseline. (In extended AIME2025 runs conducted after submission, we obtain a 2.5x end-to-end speedup with synchronous level accuracy)
>
>
> |  | Regular Backward Pass | OPOB (Single Backward) | Naïve (Double Backward) |
> | --- | --- | --- | --- |
> | Time (s) | 69.1 | 82.5 | 141.2 |
> | Peak Memory (GB) | 48.3 | 55.1 | 48.3 |
>
> ### 4. Is ESS scaling or OPOB more essential?
>
> Thank you for this insightful question. We agree that, historically, optimal baselines in RL have been impractical as they require an additional backward pass, which is why prior works default to simple mean-return baselines. However, **a key systems contribution of our work (Section 3.4) is eliminating this exact bottleneck,** making OPOB practical enough with only a 19% update-time overhead (Figure 2).
>
> Furthermore, VCPO’s components play complementary roles in variance control:
> 1. **ESS-Guided Scaling is a global control:** it controls the overall update magnitude when the effective batch size collapses to dampen unreliable high-variance updates and KL spikes
> 2. **Off-Policy Optimal Baseline enhances the gradient signal:** it reduces the variance of the gradient estimator within each update by optimally centering the reward/advantage signal using importance weights and gradient magnitudes
>
> We also run isolated ablations on the long-horizon, tool-use AIME2025 task at $k=6$
>
> | **Method Component(s)** | **Peak Val. Acc.** | **Final Val. Acc.** | **Final KL Divergence** |
> | --- | --- | --- | --- |
> | TIS | 20% | 0% (Collapsed @ Step 120) | 0.38 |
> | TIS + OPOB | 27% | 22% | 0.0052 |
> | TIS + ESS Scaling | 24% | 21% | **0.0042** |
> | TIS + VCPO (Both) | **28%** | **24%** | 0.0047 |
>
> Each one alone substantially improves stability relative to TIS, but using both gives the best final validation performance with the stability (low KL).

---

> > ### Author Rebuttal · Reviewer_pSXS · 2026-04-03
> >
> > My concerns are well addressed, and I will increase my score.

---

### Decision · Program_Chairs · 2026-04-30

**Decision:**

Accept (regular)

**Comment:**

This paper studies the high variance problem in asynchronous RL methods for training LLMs with verfiable rewards. Two approaches are proposed. One is to adaptively scale the learning rate with effective sample size and another is to use an closed-form minimum variance baseline. Both approaches are well motivated theoretically and the empirical results are convincing. All reviewers are positive about the submission and I, therefore, recommend Accept.

That being said, the biggest concern is that [x] is not discussed. Particularly, Theorem 2 of [x] gives the same optimal baseline as Eq (6) in this paper. In fact it is hard for me to believe that such a simple result has not been developed before. But I am glad to see that the authors did not put a strong claim on the novelty of this result and I acknowledge that the most insightful part of the work regarding the baseline is Section 3.4. Nevertheless, [x] should be sufficiently discussed in the final version for correct credit assignment.

[x] Zhao, T., Hachiya, H., Tangkaratt, V., Morimoto, J., & Sugiyama, M. (2013). Efficient sample reuse in policy gradients with parameter-based exploration. Neural computation, 25(6), 1512-1547.